# Aboveground and belowground sizes are aligned in the unified spectrum of plant form and function

Eleonora Beccari ● ✉ & Carlos P. Carmona ●

Understanding the global variation of plant strategies is essential for unravelling eco-evolutionary processes and ecosystem functions. Variation in ten fundamental aboveground and fine-root traits is summarised in four dimensions, the first of which relates to aboveground plant size. However, there is no consensus about how root size fits within this scheme. Here, we add rooting depth and lateral spread, compiling a set of twelve key traits that define the fundamental investments of plants in growth, reproduction, and survival. We examine whether the inclusion of root size alters the dimensionality and structure of trait correlations defining plant functional strategies. Our results show that including root size traits does not alter the fundamental structure and dimensionality of the plant functional space, regardless of trait completeness and phylogenetic relatedness. Plant size defines a single continuum of allometric investments at the global scale, independent from leaf and root economic strategies.

Distilling key dimensions of independent trait variation allows simplifying the wide diversity of plant forms and functions, revealing a continuum of strategies along which plants differentiate[1–4]. Ecologists have long tried to understand what are the fundamental dimensions of the plant phenotype[3–5] by creating synthetic schemes that combine multiple traits. By placing all vascular plant species in a single space, ecologists would be better equipped to address global change effects on ecosystems and their functioning.

The recent expansion of trait databases[6–8] has allowed different efforts in this research direction. A prime example of these efforts is the global spectrum of plant form and function[2] (GSPFF, Fig. 1), which encapsulates information for six key aboveground traits within two dimensions. The first dimension relates to the size of plants and their aboveground organs, capturing the dichotomy between herbaceous and woody species[1,2,9]. The second dimension portrays the leaf economics spectrum[10], which describes the trade-offs between resource acquisition and conservation in plant leaves[1,2,9] (Fig. 1). More recently, the advent of global fine-root databases[7,8], has allowed us to fill the gap in our understanding of root functions and their fundamental role in resource uptake, storage, transport, and plant anchorage[11], producing a synthesis that extensively explores root traits variation. Notably, the

root economics space[12] (RES, Fig. 1) reveals that fine root traits related to resource acquisition and symbiotic relationships also conform to a bi-dimensional framework. The first dimension of the RES illustrates a collaboration gradient, portraying the trade-off between species with thicker roots that rely on mycorrhizal partnerships for soil resource acquisition and species with finer roots that are better suited for independent resource acquisition[12]. Its second dimension reflects a conservation gradient, conceptually similar to the leaf economic spectrum[12] (Fig. 1). The separated spaces considering aboveground and fine root traits further highlighted the need for an integrated global synthesis of plants' above- and belowground components[5,11]. Carmona et al.[1] combined the traits defining the GSPFF and RES and defined a unified plant functional space (UPFS) demonstrating that four dimensions are needed to portray global plant aboveground and fine-roots strategies (Fig. 1). The main features of the UPFS reflect those of the individual GSPFF and RES planes, revealing the independence of above- and belowground economics at a global scale[1] (Fig. 1). In contrast, Weigelt et al.[9] proposed a potential coordination between the leaf economics spectrum and the conservation dimension of the RES (see also Weigelt et al.[13]). However, Bueno et al.[14] supported the integrity of the UPFS, suggesting that the divergent conclusions

Department of Botany, Institute of Ecology and Earth Sciences, University of Tartu, J. Liivi 2, Tartu, Estonia. ✉e-mail: eleonora.beccari@ut.ee

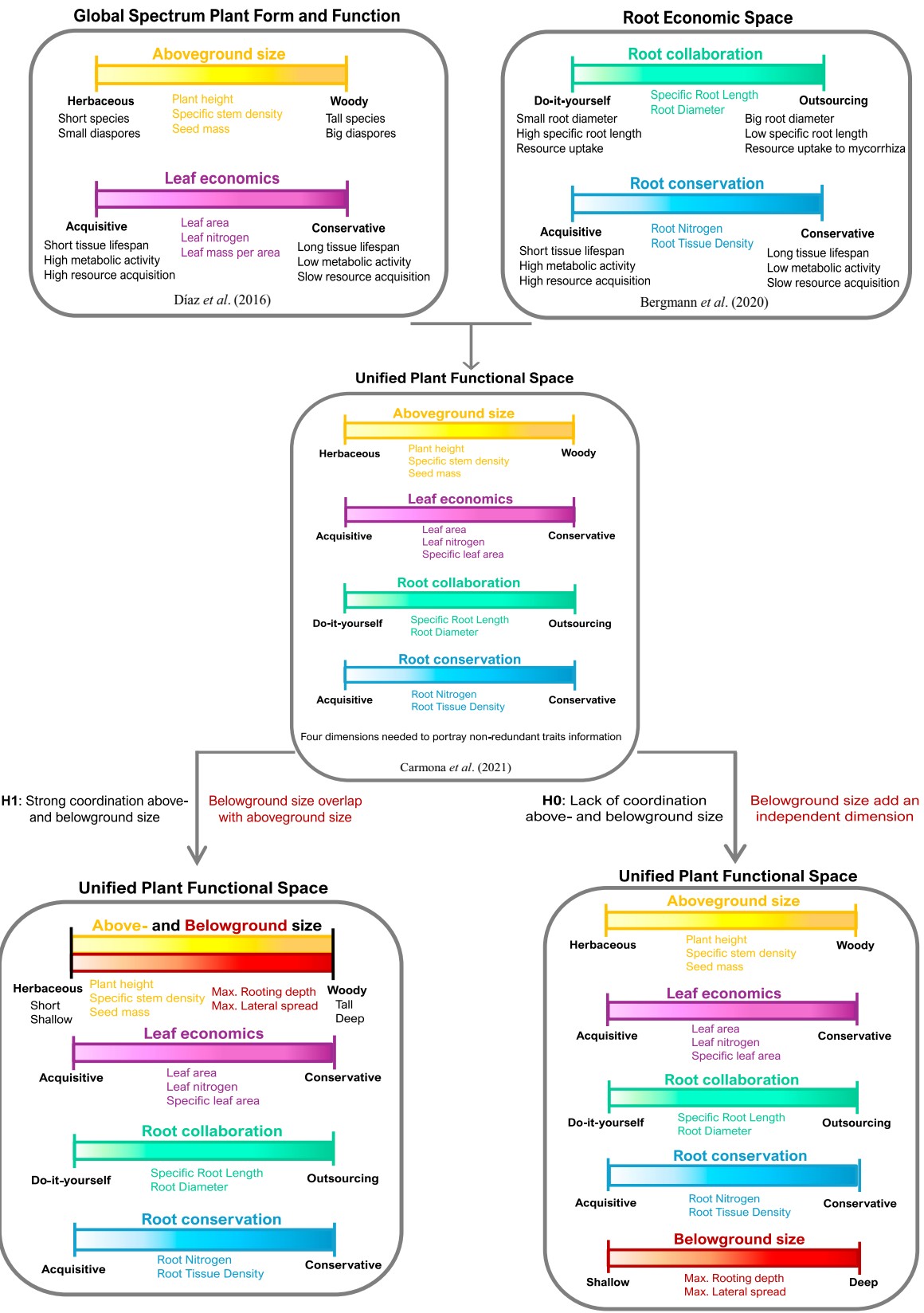

between the two studies stem from different interpretations of trait alignment in otherwise equivalent spaces.

The ongoing debate about the independence versus coordination of traits within the UPFS framework leads naturally to another critical aspect of plant functional ecology: the size of above- and belowground organs. Organ size has implications for resource acquisition (competition for light, water, and nutrient uptake)[9,15], anchorage, and plant-environment interactions[15]. Recent research supports two contrasting hypotheses regarding the coordination of above- and belowground size[9,15]. One hypothesis posits that, while there are clear benefits of increased height for light acquisition, extending the root system is not the only strategy to acquire soil resources accordingly.

**Fig. 1 | Conceptual representation of the expected effect of root size traits inclusion in the Unified Plant functional Space (UPFS).** Global Spectrum of Plant Form and Functions (GSPFF) demonstrated that aboveground plant strategies can be summarised by two independent axes of trait variation related to plant aboveground size and to leaf economics[2]. Root Economics Space (RES) showed that belowground plant strategies can be summarised along two main dimensions of belowground traits variation identified by the root conservation and root collaboration gradients[12]. A recent effort simultaneously addressing above- and belowground trait spaces (i.e., UPFS) showed a decoupling of above and belowground strategies, demonstrating that four dimensions are needed to summarise

plants' investments and that these dimensions corresponded to the one defining GSPFF and in RES[1]. The inclusion of root size traits in the UPFS can lead to two possible results. As we hypothesise, if above- and belowground size is coordinated at the global scale, root size traits will align along the aboveground size axis. Conversely, if above and belowground dimensions are decoupled, belowground size will add an independent dimension to the UPFS[9]. Each bar represents an independent dimension of the functional spaces. Traits defining each independent dimension are in the same colours of the dimension they load on. In bold black are shown the opposite plant strategies portrayed along each dimension.

Tall plants need water and nutrients to meet transpiration and photosynthetic demands; yet, deeper roots do not ensure enhanced acquisition of these resources, particularly under conditions of sufficient hydration or superficial nutrient prevalence[9]. Accordingly, there is no reason to expect coordination between above- and belowground sizes. This hypothesis is supported by Weigelt et al.[9], who found only weak correlations between height and maximum rooting depth, and described that rooting depth and aboveground size are independent dimensions in the space combining above- and belowground traits (Fig. 1). Contrastingly, allometric scaling predicts strong relationships in biomass partitioning across plant organs[16–19]. Physiological and geometrical requirements of extensive aboveground organs require balanced investments in belowground ones to support it, implying a positive coordination among above- and belowground investments in organ dimensions. Recent findings showing strong positive correlations between above- and belowground dimensions confirmed this hypothesis[15] (Fig. 1). This dichotomy of findings illustrates that the question of size coordination between above- and belowground components remains unresolved, complicating our understanding of how the allocation of resource to growth above- and belowground influences the UPFS[20].

Here, we aim to answer the unresolved questions surrounding size coordination within the UPFS. For this, we obtain information for maximum rooting depth (Dr) and maximum lateral root spread (Lr) from the database of Tumber-Dávila et al.[15], and combine it with information on the set of ten traits defining the UPFS[1], which includes plant height (ph), specific stem density (ssd), seed mass (sm), leaf area (la), leaf nitrogen concentration (ln), specific leaf area (sla), specific root length (SRL), root diameter (D), root tissue density (RTD) and root nitrogen concentration (RN). Accordingly, we assemble a dataset encompassing 39,334 plant species with different levels of trait completeness (Supplementary Fig. 1), ranging from a maximum of 23,003 species with data for seed mass to a minimum of 1230 species for root nitrogen concentration (Supplementary Table 1a). 134 species had complete observations for all twelve traits, whereas 2965 plant species had observations for at least one root size trait (Supplementary Table 1b). We use this dataset to test whether incorporating root size traits alters the UPFS dimensionality and structure of trait correlations (Supplementary Fig. 1). Given the strong correlations observed between size traits[15], we hypothesise that above- and belowground size traits will align along a single dimension of variation, so that the addition of belowground size traits will not increase the dimensionality or fundamental structure of the UPFS[1] (Fig. 1). To validate this hypothesis, we compare the structure and effective number of dimensions (END) of the UPFS to three different spaces progressively including root size data (Supplementary Fig. 1). Additionally, using the same approach, we examine the impact of root size traits on the GSPFF and the RES independently. We expect that, if above- and belowground sizes are strongly correlated, root size will not increase the dimensionality or modify the structure of trait relationships of the GSPFF. Conversely, because we expect belowground size to be fundamentally decoupled from fine root traits, we predict that, when added to the RES, belowground size will emerge as a new, independent dimension. Finally, we explore whether our findings are influenced by the

completeness of trait information and by the phylogenetic relatedness of species (Supplementary Fig. 1).

## Results

### Root size traits in the UPFS

Root size traits (Lr and Dr) were highly and positively correlated to each other and with aboveground size. Specifically, lateral root spread showed higher correlations with aboveground size ($r_{ph} = 0.71$; $r_{ssd} = 0.64$; $r_{sm} = 0.49$) compared to maximum rooting depth ($r_{ph} = 0.54$; $r_{ssd} = 0.52$; $r_{sm} = 0.44$). See Supplementary Fig. 2 for information about the correlations between all pairs of traits.

The reduced space (i.e., the space keeping only dimensions with eigenvalue > 1) built considering UPFS and root size traits included four relevant dimensions, altogether explaining 72.31% of total trait variation (Fig. 2a-f, Supplementary Table 2). The first dimension (32.14% of total variance) reflected the positive correlations observed among above- and belowground size traits. Second and third dimensions (respectively 16.18% and 14.72% of total variance) reflected the trade-off between specific root length and root diameter as well as the covariation among traits related to the leaf economics spectrum (specific leaf area and leaf nitrogen). The fourth dimension (9.26% of total variance) portrayed the trade-off between root nitrogen and root tissue density. Angles between traits in the reduced space representing the same functional dimensions (i.e., above- and belowground size, leaf economic spectrum, root collaboration, root conservation) were consistently either close to 0 degrees (showing positive covariation) or close to 180 (showing trade-offs, Fig. 3; Supplementary Table 2). Specifically, angles between above- and belowground size traits were consistently low, ranging from 14.1° between plant height and lateral root spread (r = 0.95) to 25.7° between specific stem density and rooting depth (r = 0.64; Fig. 3).

Results remained consistent when root size traits were progressively included into the UPFS (Supplementary Table 2). The original UPFS (without root size traits) had 5.67 equivalent dimensions (END), and including root size traits increased END by much less than expected under the inclusion of an uncorrelated trait ($END_{Random} = 6.49$, Fig. 4): 2.02% increase when only lateral root spread was added ($END_{UPFS\ and\ Lr} = 5.68$), 24.40% increase when only rooting depth was added ($END_{UPFS\ and\ Dr} = 5.87$), and 12.65% increase when both root size traits were added ($END_{UPFS\ and\ Dr+Lr} = 5.77$). These results indicate that most of the variation introduced by root size in our dataset is already accounted by the UPFS traits. Progressive inclusion of root size did not alter the fundamental structure of the reduced spaces, as indicated by Pearson's correlation coefficients among angles between pairs of traits in the reduced spaces close to 1 (Fig. 5a-c). Effectively, this means that the fundamental relationships between traits in the reduced spaces are not influenced by the inclusion of root-size traits.

Results were consistent within woody and herbaceous species subsets (Supplementary Tables 3-4). The progressive inclusion of root size traits did not alter the main trade-offs shaping woody and herbaceous UPFS (Supplementary Fig. 3). The relationships between above- and belowground size traits in the reduced space showed

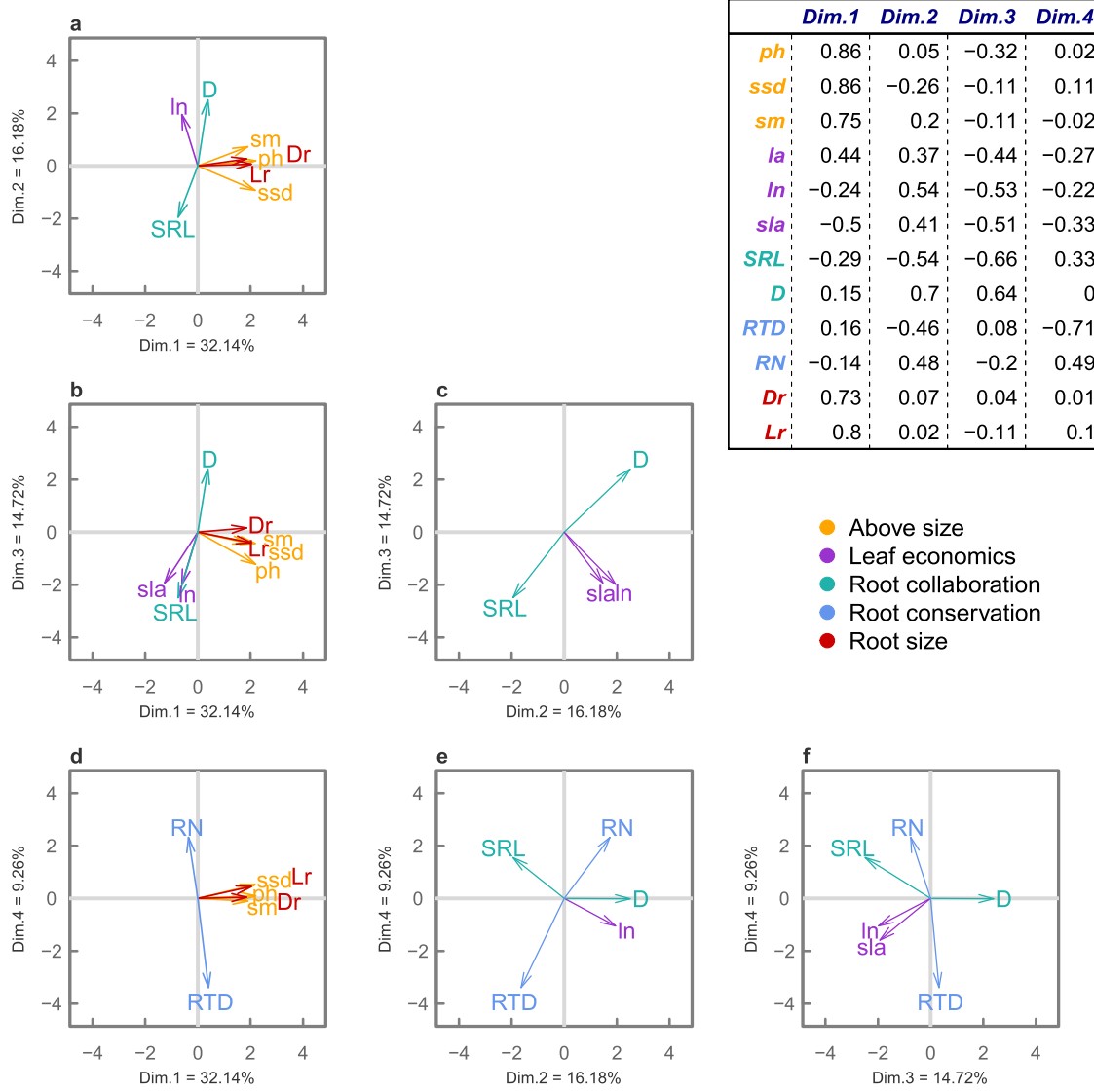

| | Dim.1 | Dim.2 | Dim.3 | Dim.4 |
|---|---|---|---|---|
| ph | 0.86 | 0.05 | −0.32 | 0.02 |
| ssd | 0.86 | −0.26 | −0.11 | 0.11 |
| sm | 0.75 | 0.2 | −0.11 | −0.02 |
| la | 0.44 | 0.37 | −0.44 | −0.27 |
| ln | −0.24 | 0.54 | −0.53 | −0.22 |
| sla | −0.5 | 0.41 | −0.51 | −0.33 |
| SRL | −0.29 | −0.54 | −0.66 | 0.33 |
| D | 0.15 | 0.7 | 0.64 | 0 |
| RTD | 0.16 | −0.46 | 0.08 | −0.71 |
| RN | −0.14 | 0.48 | −0.2 | 0.49 |
| Dr | 0.73 | 0.07 | 0.04 | 0.01 |
| Lr | 0.8 | 0.02 | −0.11 | 0.1 |

- ● Above size
- ● Leaf economics
- ● Root collaboration
- ● Root conservation
- ● Root size

**Fig. 2 | Reduced spaces built considering UPFS traits (i.e., same traits in Carmona et al.[1]) and root size traits.** Each panel shows a plane built considering pairwise dimensions: panel (**a**) first and second dimension; (**b**) first and third dimension; (**c**) second and third dimension; (**d**) first and fourth dimension; (**e**) second and fourth dimension; (**f**) third and fourth dimension. For each relevant dimension, the percentage of variance explained is shown along the axis. Only traits with loadings of at least 0.4 on either dimension are shown in each panel. Arrow length is proportional to the loadings of considered traits. Colours represent the set of traits defining the main dimension of the UPFS: aboveground size traits (orange); leaf economics spectrum (purple); collaboration gradient (light green); and root conservation spectrum (light blue). Root size traits are shown in red. The table in the upper part of the figure shows traits loadings for the four relevant dimensions. ph, plant height; ssd, specific stem density; sm, seed mass; la, leaf area; ln, leaf nitrogen; sla, specific leaf area; SRL, specific root length; D, root diameter; RTD, root tissue density; RN, root nitrogen; Dr, maximum rooting depth; Lr, maximum lateral spread.

variable strength depending on plant life form, with woody species generally showing stronger relationships for lateral root spread (angle Lr-ph = 10.6°; angle Lr-sm = 26.7°) and weaker for rooting depth (ranging from 31.6° with seed mass to 51.5° with specific stem density), whereas the relationships for herbaceous species were consistently strong for both root size traits (Supplementary Tables 3–4). Both woody and herbaceous species showed lower dimensionalities than expected under the inclusion of uncorrelated traits, although the inclusion of root size traits introduced higher variation compared to the full set of species (Supplementary Fig. 4). For woody species, the inclusion of lateral root spread introduced 8.99% of the variation expected from a single uncorrelated trait, while rooting depth alone introduced the highest proportion of variance to UPFS traits (65.39%). Conversely, for herbaceous species lateral root spread introduced the highest amount of variance (52.19%) compared to rooting depth (41.58%) to UPFS traits.

## Root size traits in the GSPFF and RES

The space including GSPFF traits and both rooting depth and lateral root spread had two relevant dimensions, which together accounted for 68.04% of the total trait variation (Supplementary Table 5). These dimensions mirrored those of the original GSPFF space, with the first dimension representing above- and belowground size and the second dimension reflecting leaf economics. By contrast, adding belowground size traits added an extra relevant dimension to the RES (Supplementary Table 6). Essentially, the first dimension was the same as in the original RES, whereas the second and third dimensions were similarly contributed by root tissue density, root nitrogen, and root size traits.

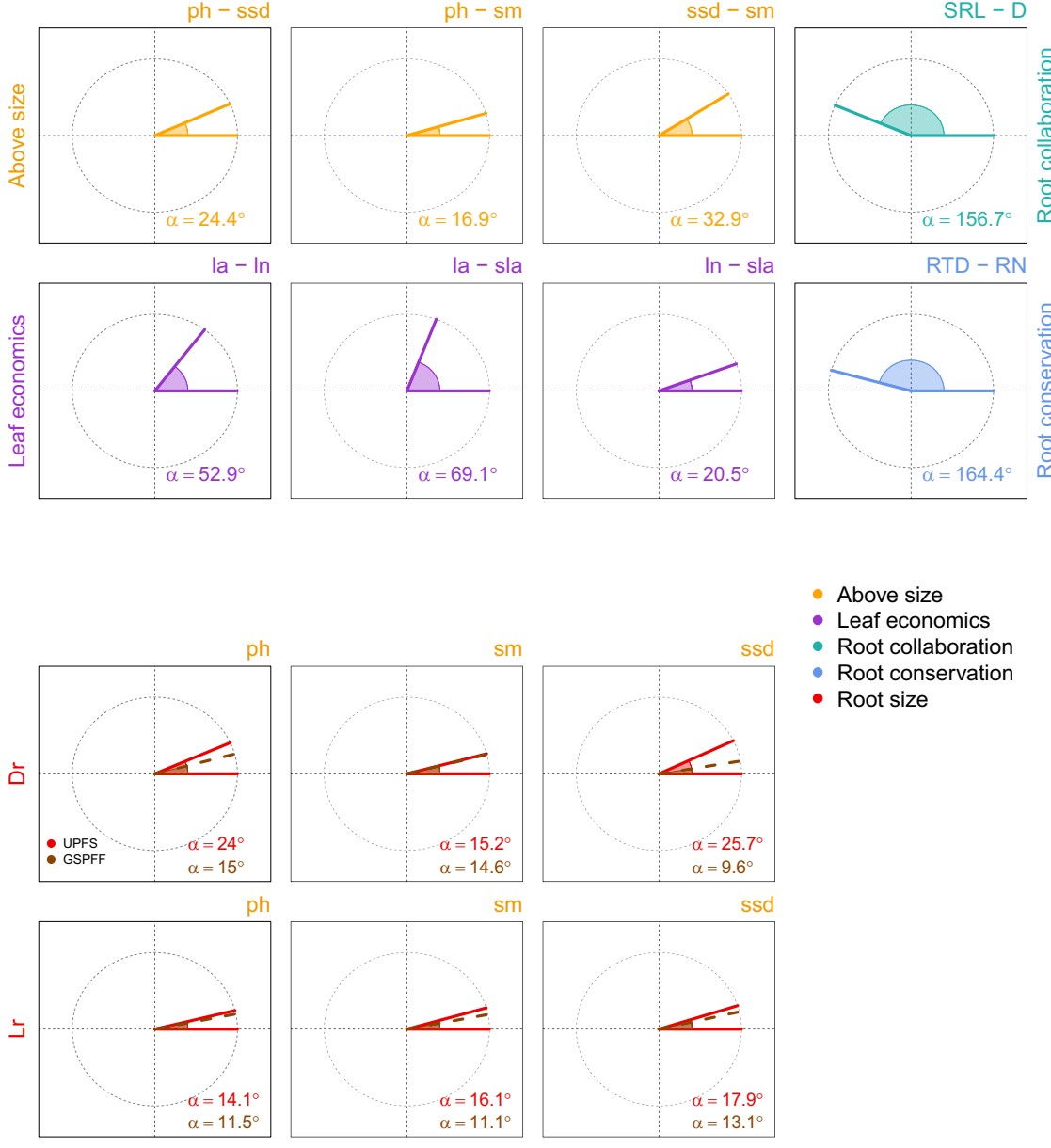

**Fig. 3 | Angles among pairs of traits in all relevant dimensions in the UPFS reduced space.** Each angle corresponds to the correlation between pairwise traits loadings across all relevant dimensions in the reduced space. Angles between 0° and 89° correspond to positive correlation coefficients, while angles between 91° and 180° correspond to negative correlation coefficients. Angles of 0° and 180° correspond to a Pearson's correlation coefficient ($r$) of +1 and −1, respectively. An angle of 90° corresponds to $r = 0$. The upper panels show the angles of trait correlations previously observed for the UPFS: aboveground size traits (orange); leaf economics spectrum (purple); collaboration gradient (light green); and root conservation spectrum (light blue). Lower panels show the angles between above- and below-ground traits. Light red angles are among above- and belowground traits in the reduced space considering UPFS traits; dark red dotted lines show angles among above-and belowground traits in the reduced space built considering GSPFF traits. ph, plant height; ssd, specific stem density; sm, seed mass; la, leaf area; ln, leaf nitrogen; sla, specific leaf area; SRL, specific root length; D, root diameter; RTD, root tissue density; RN, root nitrogen; Dr, maximum rooting depth; Lr, maximum lateral spread. All angles across each pair of traits and across the different reduced space can be found in Supplementary Fig. 2 and Supplementary Table 2.

However, inspection of the angles between pairs of traits further revealed that rooting depth and lateral root spread were strongly aligned ($\alpha = 8.81°$, $r = 0.99$) but close to orthogonal to fine root traits (Supplementary Table 6).

The progressive inclusion of root size traits did not alter the GSPFF characteristics or the main trade-offs shaping the original RES dimensions (Supplementary Tables 5-6). However, when adding only a single root trait to the RES (rooting depth or lateral root spread), we found intermediate angles among root size and both specific root length and root nitrogen (around 136°, $r = -0.73$). Despite these differences, RES angles remained consistent across all reduced spaces, as

observed for GSPFF, suggesting that root size does not significantly alter trait correlations structuring the fundamental dimensions of both RES and GSPFF (Fig. 5).

GSPFF spaces including root size showed much lower END than spaces including a random uncorrelated trait (Fig. 4), showing that most of root size variation is already accounted for by GSPFF traits. Conversely, inclusion of root size traits in RES approximated, or even exceeded, the END expected under a random uncorrelated trait ($END_{Random} = 3.77$). While the baseline dimensionality of the RES was 2.84, including individual root size traits increased it as much as 95.68% ($END_{RES + Lr} = 3.73$) and 89.13% ($END_{RES + Dr} = 3.67$) of the

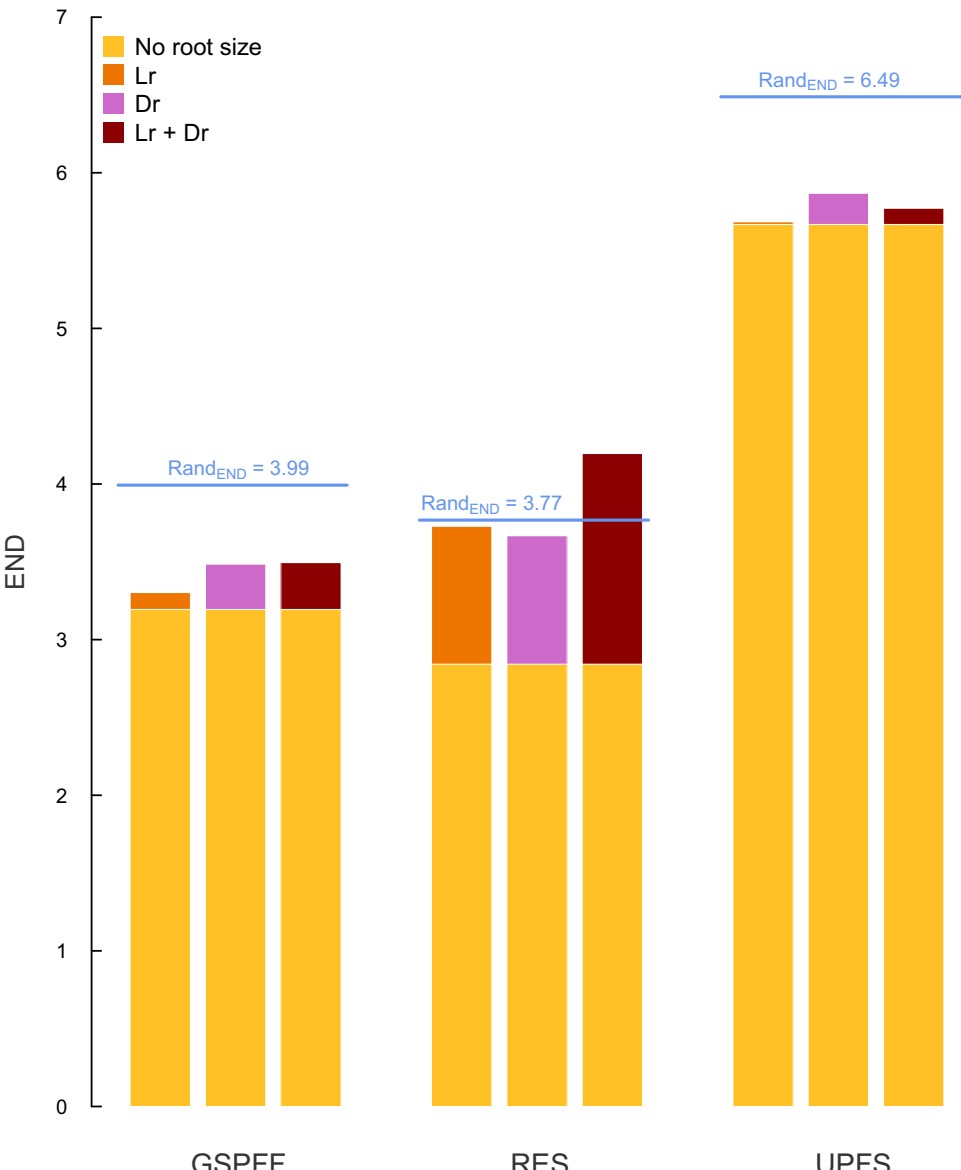

**Fig. 4 | Effective number of dimensions (END) across GSPFF, RES, and UPFS traits subsets with different levels of root size inclusion.** *No root size*, spaces excluding root size traits (yellow); *Lr*, spaces considering only lateral root spread (orange); *Dr*, spaces considering only rooting depth (pink); *Lr + Dr*, spaces considering both root size traits (dark red). Blue lines represent the END expected if we add a random uncorrelated trait to the GSPFF, RES, and UPFS set of traits ($Rand_{END}$).

increase of a random uncorrelated trait. The inclusion of both root size traits increased the dimensionality by 146.21% ($END_{RES\ and\ Dr+Lr} = 4.20$). Altogether, these results suggest that the RES contains very little information about root size, in contrast with the GSPFF.

Trait relationships in the reduced GSPFF and RES spaces were consistent with the UPFS ones across all trait subsets, with trait angles showing significant and positive correlations independent of the subsets of root size traits considered (Supplementary Fig. 5). The END of the GSPFF and the RES were 2.47 and 2.83 units smaller than that of the UPFS, respectively (Fig. 4). The inclusion of root size traits did not substantially modify these differences for the GSPFF (on average, 2.35 units smaller than UPFS), but decreased them in the case of the RES (on average, 1.91).

**Testing the consistency of the UPFS, GSPFF, and RES spaces**
The structure and dimensionality of PCA spaces based on species with complete trait information were consistent with those of the UPFS,

GSPFF, and RES (Supplementary Tables 7-9). Accordingly, we found that correlations among trait angles were close to 1 across all reduced spaces, confirming that trait correlations observed in the main reduced space are not influenced by the completeness of trait observations (Supplementary Tables 10-11). END showed the same trends observed in the main analyses across all trait subsets (Supplementary Fig. 6), with the only exception being the UPFS, where spaces including root size traits had even lower dimensionality than the one excluding root size.

Results observed in the main analyses remained consistent when accounting for phylogeny across all UPFS, GSPFF, and RES trait subsets (Supplementary Tables 12–14). We found strong and positive correlations among trait angles of all phylogenetically informed reduced spaces, with the only exception of RES reduced spaces including single root size traits (Supplementary Table 10). We also found strong and positive correlations among trait angles in all phylogenetically-informed reduced spaces and those from the main analyses,

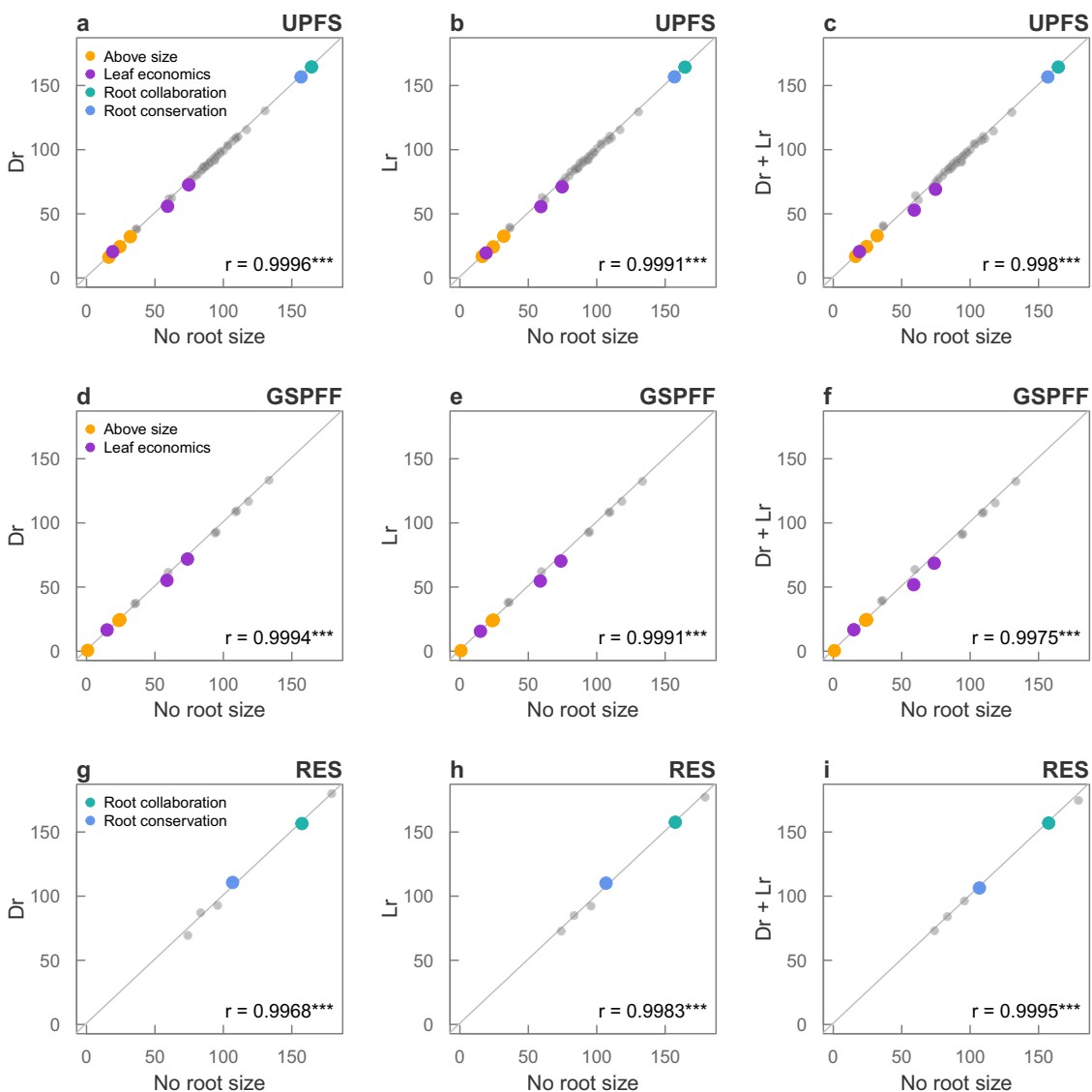

**Fig. 5 | Correlations among angles in the reduced spaces built progressively including root size traits across UPFS, GSPFF and RES trait subsets.** *No root size*, trait subsets excluding root size traits; *Dr*, trait subsets considering only rooting depth; *Lr*, trait subsets considering only lateral root spread; *Dr + Lr*, trait subsets considering both root size traits. Each dot represents an angle among pairwise variables. Coloured dots are the angles among highly correlated variables defining the main trade-offs present across the UPFS, GSPFF, and RES: aboveground size traits (orange); leaf economics spectrum (purple); collaboration gradient (light green); and root conservation spectrum (light blue). Correlation coefficients (r) are shown in the lower part of each panel, along with their two-sided significance (***, $p$-value < 0.001). Exact $p$-value for each panel: **a** = $1.40e^{-67}$; **b** = $3.13e^{-60}$; **c** = $5.05e^{-53}$; **d** = $2.24e^{-20}$; **e** = $3.31e^{-19}$; **f** = $2.46e^{-16}$; **g** = $1.55e^{-05}$; **h** = $4.52e^{-06}$; **i** = $4.11e^{-07}$.

suggesting that phylogenetic history does not significantly impact trait correlations in the reduced spaces (Supplementary Table 11). Changes in END values for the phylogenetically-informed spaces after adding root size traits showed generally the same patterns observed in the main analyses (Supplementary Fig. 6). Interestingly, END of the phylogenetically-informed spaces were generally higher than what was observed in the main analyses: phylogenetically-informed UPFS showed an average increase of 0.86 dimensions, GSPFF of 1.42, and RES of 0.05 dimensions.

## Discussion

The recent integration of the global spectrum of plant form and function[2] and the root economics space[12] showed that these can be summarised in a four-dimensional space where aboveground and fine root strategies of plants are independent[1] (UPFS, Fig. 1). Yet, it remained unclear how root size would fit into this scheme[9,13–15]. By integrating maximum rooting depth and lateral root spread to the

UPFS framework, we demonstrate that (1) above- and belowground plant sizes are strongly aligned, forming a single dimension of variation; and consequently, (2) the inclusion of root size traits does not impact the dimensionality and fundamental structure of the UPFS.

### Above- and belowground size are aligned in the UPFS

Our results show that global variation of belowground size is already described along the dimension retaining aboveground size variation in the GSPFF (Figs. 2–5), which corresponds to the expectations set by allometric relationships[17,21] and a recent global synthesis[15]. Above- and belowground sizes were strongly correlated in all considered spaces (Fig. 3, Supplementary Tables 2,5–6), so that root size variation found at the global scale loaded only on the first dimension of the spaces, regardless of the subset of root size traits considered. As a result, the inclusion of both root size traits in the GSPFF and the UPFS changed the dimensionality of the spaces to a much smaller extent than expected by the inclusion of a single uncorrelated trait (Fig. 4). These

results empirically suggest that, at the global scale, variation in root size is already explained by the aboveground size dimension of these spaces and confirm this dimension as an indicator of overall plant size.

Physiological and geometric constraints imposed by body dimensions impose an allometric scaling in how plants allocate resources between above- and belowground organs[17,18,21,22]. Accordingly, biomass aboveground is predicted to be proportional to that belowground[17,21], a pattern observed regardless of the phylogenetic affiliation of species or environmental pressures[17,18,23,24]. This proportional relationship stems from the fact that larger plants generally need extensive root systems to satisfy their nutrient and water needs, as well as to anchor aboveground organs. These expansive root systems, in turn, demand significant energy for maintenance, which requires a large photosynthetic area[22]. Accordingly, when the global spectrum of plant form and function is considered[1,2], there is a strong coordination in above- and belowground size investments[15,20,22]. This coordination is evident along the first dimension of independent trait variation in the UPFS. However, it is important to acknowledge that despite the high accessibility and prevalent use of plant height as a proxy for aboveground plant biomass[1,2,9,10,25], plant height may not be the optimal proxy for herbaceous species[26], particularly those with morphologies that do not detach significantly from the ground (e.g., rosettes or tillers). In such cases, shoot biomass is suggested to provide a more accurate representation of aboveground investments[26].

## Belowground size does not impact the fundamental structure of the UPFS

Stressing the strength of the coordination of size investments, the inclusion of root size traits did not modify the fundamental structures of the GSPFF and UPFS (Fig. 5). This finding underscores a key aspect of plant trait dimensionality: while aboveground and fine-root traits related to construction costs and resource acquisition differ at a global scale[1], traits defining plant size demonstrate a remarkable consistency. Our analyses show that root trait variation is summarised along three independent dimensions of variation, such that adding either root size trait to the RES expands the dimensionality of this space to a similar extent as adding an uncorrelated trait. Interestingly, this aspect aligns with the results of Weigelt et al.[9], but with a notable divergence. Unlike their study, which found only a weak correlation between aboveground size and rooting depth, our analyses indicate a stronger linkage. We attribute the observed disparity to the imbalance in the representation of woody species between our study and that of Weigelt et al.[9]. Specifically, of the 540 species with common information for plant height and rooting depth in Weigelt et al.[9], 62.41% were woody and 33.52% herbaceous. By contrast, in our study, out of 1631 species sharing observations for the two traits, 36.54% were woody and 58.86% were herbaceous (around 45%–48% of global vascular plant species are estimated to be woody[27]). Factors such as variations in water table depth, aridity, and precipitation significantly influence the rooting depth of woody species globally[15,28]. In our dataset, when analysed independently, woody species exhibited greater variability in rooting depth compared to herbaceous species, albeit without delineating an independent axis of variation (Supplementary Fig. 4). The discrepancy in the proportion of woody species between the two studies may potentially result in amplification in the rooting depth disparities among species showing different growth forms, resulting in different dimensionality between the two spaces. Crucially, our findings demonstrate a robust plant size coordination in the UPFS, unaffected by the specific root size traits or considered growth form, the completeness of trait information, or the phylogenetic relationships of the species. These results point to a strong, consistent coordination between above- and belowground size traits in the global context of the UPFS. Nevertheless, we advocate for future research to focus on differential investments within woody and herbaceous growth forms and to examine the impact that varying

proportions of these growth forms have on the outcomes of global syntheses like ours.

While the main findings remained consistent, spaces informed by phylogenetic data generally showed a higher dimensionality, averaging an increase of one END unit. However, regardless of the inclusion of phylogenetic information, the dimensionality and fundamental structure of the RES remained unchanged, even when root size traits were added. This outcome suggests the phylogenetic independence of plant investments along the root collaboration and conservation gradients defining RES dimensions (Fig. 1). Conversely, the higher dimensionality found in phylogenetically-informed spaces built on GSPFF traits suggests phylogenetic coordination among aboveground dimensions (Fig. 1), with the positioning of species along plant size (considering both above- and belowground sizes) and leaf economics dimensions appearing to be more closely tied to the evolutionary history of species[29] (Supplementary Fig. 6). Accordingly, the evolution of above- and belowground global plant strategies may have been subject to different selection pressures[1] (e.g., differences in resource distribution among above- and belowground components[11]), potentially leading to a decoupled evolutionary trajectory in belowground but not in aboveground investments. It is important to note that phylogenetically-informed methods are based on specific assumptions (e.g., specific models of evolution), which require critical evaluation[30,31]. Therefore, we advocate for future studies exploring phylogenetic influence in defining plant aboveground and belowground investments and the coordination among them. Despite these evolutionary variations, the relationships among traits forming the UPFS main phylogenetically-independent axes remained consistent with those in the non-phylogenetically corrected space. This underscores the pervasiveness of global trait correlations and trade-offs across plants, highlighting their fundamental role in shaping plant strategies on a global scale.

## Rooting depth and the limits of a global traits-based approach

Incorporating root size traits into the UPFS revealed nuanced differences between lateral root spread and rooting depth. Lateral root spread exhibited consistently stronger correlations with aboveground plant size than rooting depth, particularly for woody species (Supplementary Table 3). This finding aligns with previous research emphasising the role of extensive lateral root systems in stabilising taller plants[15,22]. When integrated into the UPFS framework, lateral root spread barely altered the dimensionality, whereas rooting depth contributed to a slightly larger increase in dimensionality (Fig. 4). Conversely, the variability introduced by rooting depth could stem from a more complex interaction between aboveground size and rooting depth, varying across different scales and plant growth forms. In this sense, theory predicts that, after accounting for the allometric relationship among plant organs and depending on species functional type[21,23], plants preferentially allocate biomass to the organ facing the greatest resource limitation at the expense of other organs[18,21,22]. For instance, woody species in arid environments often develop deeper roots to access low water tables[28,32] at the expense of lateral root spread and plant height[15]. Accordingly, rooting depth aligned less markedly with aboveground size traits in woody than in herbaceous species (Supplementary Tables 3-4). Similarly, this difference in variability among root size traits may stem from covariation with other plant investments not included in the framework. Indeed, clonality-related traits may play a role in influencing the complexity of above- and belowground size trait coordination, particularly in the UPFS framework[11,33,34]. For instance, clonal herbaceous traits (e.g., lateral spread of clonal organs), positively correlate with traits related to aboveground size[19,34], suggesting a possible role of clonality in influencing the covariation among above- and belowground biomass investments.

While we advocate for future efforts to disentangle the effects of biomes, clonal traits, and growth forms in defining UPFS trait

covariation, these discrepancies in trait variability introduced among growth forms underscore the intrinsic limitations of trait-based global approaches. More than 64% of known plant diversity lacks representation in global trait datasets[35], with trait databases strongly biased toward the global North[35,36]. Thus, as suggested above, global diversity frameworks, although fundamental[1,5,9], are strongly biased in species representation. This underscores the need to integrate under-sampled growth forms, traits, and biomes to fully understand global plant strategies and their patterns in community assembly[35–37]. In particular, to comprehensively understand the drivers behind global trait covariation and patterns in multivariate trait spaces, efforts should focus on exploring how traits covary along environmental gradients[38].

Furthermore, it is important to emphasise the inherent limitations of trait measurements. Traits that are difficult to measure may present biases in their estimation. For instance, our limited understanding of root size[15] is primarily due to the challenges in extracting measurements for the deepest-rooted species, leading to a general underestimation of global root size[28]. Nevertheless, with the most up-to-date datasets, our global analysis indicates a strong, overarching alignment of above- and belowground size traits that is independent of growth form and phylogeny. This suggests a unified continuum of size-related traits across the plant kingdom, distinct and independent from leaf and root economic strategies.

## Methods
### Data collection and processing
We used information for the six aboveground traits that compose the GSPFF[2] and the four fine-root traits that compose the RES[12] (Fig. 1). The GSPFF trait information was derived from the TRY database[6,39], whereas the RES traits were downloaded from the GRooT database[8] (Supplementary Fig. 1). To create the UPFS, we assembled the TRY and GRooT datasets using the same trait subsets and the same data processing described in Carmona et al.[1]. Specifically, the UPFS dataset included ten measured traits comprising: plant height (ph, measured in m), specific stem density (ssd, g/m³), seed mass (sm, mg), leaf area (la, mm²), leaf nitrogen concentration (ln, mg/g), specific leaf area (sla, mm²/mg), specific root length (SRL, m/g), root diameter (D, mm), root tissue density (RTD, g/cm³) and root nitrogen concentration (RN, mg/g, Supplementary Fig. 1).

We collected the two root size traits from the dataset used in Tumber-Dávila et al.[15], which included over 5600 observations for 2989 species across different geographic and climatic distributions. We extracted information about maximum rooting depth (Dr, measured in m), and maximum lateral root spread (Lr, one-sided, m). We excluded all observations without a clear taxonomic classification or with higher taxonomic ranks than species (i.e., families, or genus). Then, we averaged trait records for all species with multiple observations, producing a root size dataset of 2965 plant species (Supplementary Fig. 1).

We combined all traits information (UPFS and root size), producing a final dataset of 39,334 plant species and twelve traits with different levels of completeness (Supplementary Fig. 1, Supplementary Table 1a). Observations were log-transformed and scaled for subsequent analyses; outliers were set as NAs. Nomenclatures of species across the dataset were standardised following World Flora Online taxonomy[40] using the *WFO.match* function of the R package 'WorldFlora'[41]. We resolved fuzzy classifications using the *TPL* function of the R package 'Taxonstand'[42], manually checking fuzzy records on World Flora Online[43].

Considering the full set of 39,334 species, pairs of traits with common observations ranged from a minimum of 257 common species (for root nitrogen and lateral root spread) to a maximum of 7217 (plant height and seed mass, Supplementary Table 1a). The traits with the highest completeness were seed mass (N = 23,003, observations available for the 58.48% of total species) and plant height (N = 13,438,

34.16% of total species), whereas the least complete traits were root nitrogen concentration (N = 1230, 3.13% of total species) and root tissue density (N = 1361, 3.46% of total species; Supplementary Table 1a). The dataset contained 49.47% of woody species and 32.04% herbaceous species (with 18.29% of species missing woodiness information). See Supplementary Table 1a for information about trait completeness.

Finally, we downloaded phylogenetic information for all considered species using the *phylo.maker* function of the R package 'V.PhyloMaker2'[44]. Specifically, we used the *GBOTB.extended.TPL* phylogeny, which originally considers 74,531 species, and we used scenario S3 to extract the phylogenetic information for our considered subset of species[45] (Supplementary Fig. 1).

### Root size in the UPFS
We calculated Pearson correlation coefficients across the full set of twelve traits (Supplementary Fig. 2). Then, we tested whether the inclusion of root size traits affects dimensionality and the correlation structures defining the main dimensions of trait variation observed in the original UPFS (i.e., trait correlations along the four main axes of trait variation of Carmona et al.[1]). For that, we performed a series of four eigendecompositions progressively including root size traits in the set of ten traits used to portray the UPFS[1]. Specifically, we considered: 1) UPFS traits (above- and belowground traits, mirroring Carmona et al.[1], 2) UPFS traits and rooting depth (above- and belowground traits and Dr), 3) UPFS traits and lateral root spread (above- and belowground traits and Lr), 4) UPFS and root size traits (above- and belowground traits with both Dr and Lr, Supplementary Fig. 1, Supplementary Table 2). Eigen-decomposing a matrix of trait correlations is equivalent to performing a principal component analysis (PCA) on trait correlations. Yet, unlike a standard PCA, the eigendecomposition approach allows the inclusion of observations (species) with missing trait values (because the correlations between pairs of traits consider the set of species that have common information for those two traits[1,12]), thus reducing the bias introduced by considering only species with complete trait information, which are a non-random subset of species[46]. This approach allowed us to extract relevant dimensions of non-redundant trait variation while maximising the information used to estimate each pairwise correlation. Using this procedure, we estimated a correlation matrix for each of the four sets of traits described above (Supplementary Fig. 1). After performing an eigendecomposition of these correlation matrices, we calculated the number of relevant dimensions needed to portray non-redundant trait variation by retaining dimensions with eigenvalues larger than 1[47]. For the sake of simplicity, we will hereafter refer to spaces that encompass only the relevant dimensions of trait variation as 'reduced spaces', whereas, when all dimensions of trait variation are considered, we will address them simply as 'spaces'.

For each eigendecomposition, we estimated trait loadings, which reflect the correlations between traits and relevant dimensions of trait variation[48]. Then, to understand the relationships among traits in each reduced space, we estimated the correlations between their loadings considering all relevant dimensions in the space (Supplementary Fig. 1). Following Bueno et al.[14], we expressed these correlations as the angles that each pair of traits forms in the reduced space; this procedure allows for graphically representing the relationships between traits, easing interpretations of trait-trait relationships in the reduced space[1,14] (Supplementary Table 2). Finally, we compared the dimensionality and trait correlations in each of the four spaces built by progressively including root size traits (Supplementary Fig. 1).

While retaining dimensions with eigenvalues >1 is a common approximation for determining the dimensionality of the reduced space[49,50], this binary approach is too coarse to fully understand how the addition/removal of any given trait affects the space. For example, adding a trait that is completely uncorrelated with the other traits should result in an increase of one dimension, whereas adding a trait

that is perfectly correlated with any other trait should not modify the dimensionality at all. However, in most real scenarios, the added trait has intermediate correlations with the other traits. To get a more sensitive and continuous estimation of the effects of trait addition/removal, we calculated the effective number of dimensions (END) for each space (Supplementary Table 2). For that, we estimated the inverse of the Simpson's diversity index based on the eigenvalues of each space[51–53], using the *diversity* function of the R package 'vegan'[54]. Inverse Simpson is a standardised measure of diversity reflecting the number of equally abundant species needed to obtain the observed proportional abundances in the sample[51–53]. Accordingly, by substituting species with dimensions of trait variation, and abundances with the variance explained, we produced a standardised and continuous metric of dimensionality (Supplementary Fig. 1). END represents the dimensionality of a set of traits, encapsulating the extremes of perfect correlation (END = 1) and complete lack of correlation among all traits (END = the number of traits used to build the space).

We made a null model to compare the increase in END observed after adding root size traits with the change in dimensionality expected if an uncorrelated trait were added to the UPFS. For that, we simulated a random uncorrelated trait by sampling from a random normal distribution with a mean of 0 and a standard deviation of 1, we added this simulated random trait to the subset of traits defining the UPFS (excluding root size traits), we performed an eigendecomposition, and we estimated END of the resulting space. We repeated the same procedure 500 times to obtain a null distribution of END. Then, we used the mean END of the simulated space as a reference to quantify to what extent the inclusion of root size approximates the inclusion of an uncorrelated trait in the UPFS (Fig. 4). For this, we calculated the difference between the observed END of the UPFS and the simulated one. Then, we used this difference as a reference to calculate the proportional dimensionality increase (expressed as a percentage) for the three spaces including root size traits compared to the UPFS without root size traits. Values of proportional increase approaching 0 would indicate that the inclusion of root size traits does not change the dimensionality of the space, and vice versa.

Finally, to test whether the inclusion of root size traits influences trait correlations in the reduced spaces, we compared the trait angles across the four reduced spaces (Supplementary Fig. 1). For that, we estimated the correlations among pairs of angles in the UPFS reduced space with those in the reduced spaces progressively including root size traits. If the angles in the reduced space including root size traits are strongly and positively correlated with the angles estimated on reduced space not including root size traits, it can be concluded that the inclusion of root size traits does not modify the fundamental characteristics of the reduced space (i.e., the relationships between other traits)[1,14].

Additionally, aiming to test whether observed patterns in the UPFS were maintained independently on species woodiness, we performed the same series of eigendecompositions within woody and herbaceous species subsets. We extracted woodiness information from Carmona et al.[1], providing woodiness classification for 32,060 species (c.a. 81.51% of the total dataset; Supplementary Table 1a). Then, we divided species with woodiness classification into two datasets (i.e., woody and herbaceous datasets), excluding species without woodiness information. For each dataset we extracted the correlation matrix, we performed the same series of eigendecompositions progressively including root size traits to the set of ten traits used to portray the UPFS, we extracted trait angles in the reduced spaces, and we computed END of the spaces by using the same procedure used above for the full dataset (Supplementary Tables 3–4). Then, for both woody and herbaceous reduced spaces we compared trait correlations across the four spaces built progressively including root size traits (Supplementary Fig. 3). Additionally, we simulated the inclusion of an uncorrelated trait for both woody and herbaceous

UPFS, and we computed the associated change in dimensionality (Supplementary Fig. 4).

### Root size in the GSPFF and RES
In order to understand the relationships between root size traits and only aboveground (GSPFF[2], Fig.1) and only belowground traits (RES[12], Fig.1), we performed the same set of analyses described above considering separately the subset of traits that conform the GSPFF and the RES (Supplementary Fig. 1). For each of these subsets, we performed the same set of four eigendecompositions progressively including root size traits: 1) without root size traits, 2) with Dr only, 3) with Lr only, 4) with Dr and Lr (Supplementary Tables 5-6). In short, for each eigendecomposition, we (1) extracted angles among pairs of traits in the reduced spaces and computed the END of the spaces; (2) simulated the inclusion of an uncorrelated trait as done above for the UPFS and computed the associated change in dimensionality; and (3) tested whether the inclusion of root size traits influences trait correlations in the reduced GSPFF and RES (Fig. 5, Supplementary Fig. 5).

### Testing the consistency of the UPFS, GSPFF, and RES spaces
To test whether spaces' structures remained consistent when considering only complete trait observations, we estimated the three main spaces explained above (UPFS, GSPFF, RES) considering the same levels of root size inclusion (3 spaces x 4 levels of root size inclusion = 12 trait spaces, Supplementary Fig. 1). For each of these trait spaces we removed all species with missing trait information for any of the considered traits, creating datasets with variable number of species depending on the completeness of the considered traits (see Supplementary Fig. 1 for examples and Supplementary Table 1b for trait subsets completeness). For each dataset, we repeated the same set of analyses specified above, including quantification of relevant dimensions using Kaiser's rule, computation of angles between pairs of traits in the reduced space, and comparison of these angles across levels of root size inclusion, as well as the estimation of END and null models' END (Supplementary Fig. 6, Supplementary Tables 7–10). Additionally, we correlated angles of all twelve USPS, GSPFF, and RES traits subsets with the corresponding ones computed considering the full set of species (i.e., main analyses) by extracting Pearson's correlation coefficients. High positive correlations would suggest that trait correlations observed in the main analyses are maintained when considering species with complete trait information (Supplementary Table 11).

Similarly, we tested whether relevant dimensions and trait correlations in the reduced spaces remained consistent after accounting for species phylogenetic history (Supplementary Fig. 1). For that, using species with complete trait information, we computed twelve phylogenetically-informed PCAs (PPCA[55]) using the *phyl.pca* function of the 'phytools' R package[56]. For each PPCA we performed the same analyses described in the previous paragraph for the spaces built using complete trait information. Given the time-consuming nature of the PPCA computational procedure, proportional increase in END in the PPCA spaces was determined considering a single simulation for each subset of traits (Supplementary Fig. 6, Supplementary Tables 10-14)

All analyses were performed using R version 4.0.3[57].

### Reporting summary
Further information on research design is available in the Nature Portfolio Reporting Summary linked to this article.

## Data availability
The data used in the analysis can be found on Figshare at https://doi.org/10.6084/m9.figshare.25600650. TRY data can be accessed at https://www.try-db.org/TryWeb/Home.php. Original GRooT data are available at https://doi.org/10.1111/geb.13179. Original root size data can be found at https://doi.org/10.1111/nph.18031. Data on species nomenclature are available at https://www.worldfloraonline.org. Plant

phylogenies can be accessed at https://doi.org/10.1016/j.pld.2022.05.005.

## Code availability

Data and codes required to replicate the analysis can be found in Figshare (https://doi.org/10.6084/m9.figshare.25600650).

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

## Acknowledgements

We thank Enrico Tordoni, Francesco Petruzzellis, and Lorenzo Beccari for insightful discussions on plant strategies. E.B. and C.P.C were supported by the Estonian Research Council (PSG293, PRG2142), the European Union (ERC, PLECTRUM, 101126117), and the European Regional Development Fund via the Mobilitas Pluss (MOBERC40) and Mobilitas 3.0 (MOBERC100) programmes.

## Author contributions

C.P.C. and E.B. conceived the study and designed the analyses. E.B. collected, processed, analysed the data, and wrote the first draft of the manuscript. C.P.C. and E.B. contributed to the interpretation of results and article writing.

## Competing interests

All authors declare no competing interests.
