## [Transparent Peer Review file · Nature Communications]

Aboveground and belowground sizes are aligned in the unified spectrum of plant form and function

Corresponding Author: Ms Eleonora Beccari

Version 0:

Reviewer comments:

Reviewer #1

(Remarks to the Author)

The question of the dimensionality of trait space is of major importance both in terms of pure science (e.g., evolutionary and ecological constraints on morphology) and sampling design (e.g., selecting orthogonal traits to sample). The authors set forth to test a key question relating to the dimensionality of combined above-and-belowground trait spaces. They assemble a sizable dataset (given the limited availability of trait data) and take several approaches to addressing this question. The approaches used are all reasonable, and importantly, they all point to a similar conclusion: that above- and below-ground size traits form a single spectrum.

The manuscript itself is in great shape and the message and writing very clear, thus I only have minor comments:

Figure 1 is great! It not only helps clarify some of the terminology, but also clearly sets out the main hypothesis being tested

Line 68 -83: "Lr", "Dr", etc. - since the methods are at the end in this article format, this is the first place readers will be seeing these acronyms, so it would be good to define them here as well.

Work by Uyeda et al. (<https://doi.org/10.1093/sysbio/syv019>) suggests that PPCA can be misleading (at least in some contexts). Perhaps mention this somewhere in the discussion? Given that the evolutionary aspect isn't the main focus of this manuscript, and that you get similar results either way, I think simply mentioning this is fine without the need for any new analyses.

By providing the full code and data the methods are completely transparent and reproducible. I very much appreciate the effort spent on writing and documenting the code.

Reviewer #2

(Remarks to the Author)

The authors add rooting depth and lateral spread to a larger set of traits to ask whether root size traits alters the dimensionality of plant form and function. Determining the dimensionality of plant traits is a core objective in comparative functional ecology. The authors suggest that incorporating root size traits did not change the fundamental structure and dimensionality of the plant functional space, regardless of trait completeness and phylogenetic relatedness. Plant size emerged as a single continuum, independent from leaf and root economics strategies. While I appreciate the analysis, I have several concerns.

The authors introduce the paper as if the controversy over Carmona's (the second author) last paper did not exist, which is difficult to overlook. Carmona et al. 2021 analyzed a subset of the data in this paper and concluded that leaf and root economics (nitrogen concentrations, most notably) must be independent, despite there being no mechanistic biological explanation for such an interpretation. Virtually all nitrogen in leaves must enter through the roots, so it is difficult to reconcile the model results with ecological processes. Then the authors set up two contrasting "hypotheses", one that mentions the Weigelt et al. 2021 paper (which is the paper that disagreed with the Carmona et al. 2021 paper), but that paper also included root size, so in some ways this analysis has already been done. In sum, the authors add rooting depth to this paper to see if it changes their recent results.

The authors don't adequately address the potential relationship between aboveground height and rooting depth, which would be useful for generating more ecologically interesting hypotheses. The deepest-rooted plants on earth are also quite short in height. This is because accessing deep soil water is needed in arid climates, where it is not desirable or needed to allocate carbon to tall canopies. This is mentioned as an aside toward the very end, rather than integrated into the framing of the paper. I'm not saying that there is an inevitable trade-off globally, but allometric scaling is not the most fruitful way to approach the problem. The 'allometric scaling' hypothesis also ignores the well-known wide variation in root-to-shoot ratios that exists among plants. These ecological facts are generally ignored.

It is odd that the authors only use the raw maximum depth and maximum spread, rather than incorporating the ratios or relationships, given that these were important in the New Phytologist paper that introduced the RSIP database. Could the authors see if these dimensions of rooting architecture interact with the other traits?

The paper has many unhelpful acronyms that made the paper very difficult to read: GSPFF, RES, UPFS, END, Dr, Lr, etc... It is especially not helpful to abbreviate rooting depth (Dr) and lateral spread (Lr) by using the name of these traits in the RSIP database. Just spell them out. Moreover, they aren't even defined at first mention on L 68. END is not defined at first mention L 85. These choices undermine the readability of the ms.

Reviewer #3

(Remarks to the Author)
Dear authors and Editor,

I have now read the review manuscript entitled "Above and belowground size form a single dimension in the unified spectrum of plant form and function". In this paper the authors studied how traits related to plant size, particularly of belowground plant organs (i.e., rooting depth and root lateral spread) fit into the recently established unified spectrum of plant form and function. In their study, the authors test two main contrasting hypotheses regarding the coordination of above- and belowground size: First, that there is only a weak relationship between above- and belowground sizes, as in contrast to plant height, root size may not solely explain resource acquisition belowground. Alternatively, allometric scaling results in strong relationships in biomass partitioning across plant organs (if you are big above, then also belowground), which results in a strong relationship between above- and belowground sizes. The authors present interesting findings which support the latter, demonstrating that both above- and belowground plant sizes are strongly aligned and forming a single dimension of variation. This confirms previous evidence, but now also on a whole plant perspective including both above- and belowground traits of the global spectrum of plant form and function and the root economic space. I very much enjoyed reading the manuscript and the work will be of great interest in the current debate about the dimensionality of above- and belowground form and function; the results are very well presented and the data is rigorously analysed. However, while I like the limitations paragraph in the discussion it would be better to also address potential limitations of the data, not only in terms of global coverage, but also regarding the quantification of the traits itself. The database is certainly impressive but we still have lots to do in order to better understand how deep plants can root, and why. I also felt that that the claim of an independent above- and belowground dimension of the authors could be more nuanced, given the still ongoing debate which was issued in a commentary in Nature. Further, I think some further clarification could help to the reader to better digest the methods (which are quite complex). Nevertheless, this is a thorough, well-written contribution, and I enjoyed reading the manuscript. See below my more detailed comments that might be taken into consideration:

Comments:

Line 35-37: You state that the unified plant functional space demonstrated the independence of above- and belowground economics at a global scale. Given the current debate about the independence of above- and belowground dimensions, which is still ongoing until now, I would advocate for a more nuanced view here. At least refer to the contrasting findings by Weigelt et al. 2021 (New Phytologist) and/or the commentary in Nature (Weigelt et al. 2023).

Line 41-44: It could also be possible that adult plant height is not a good proxy for resource investment aboveground, particularly in herbaceous plants which strongly vary in morphology (e.g. rosette, tillers). Shoot biomass could be better suited to characterize the plant size gradient at least for herbaceous plants; see Funk et al. 2024, Functional Ecology.

Line 52-54: What does this number tell me in this context? I doubt that you have complete trait information for even a small fraction of these species so this number sounds like an overstatement here. I would at least expect some more detailed information about the size of complete cases when you include belowground plant size traits.

Line 64-65: That might have a huge effect on your analyses, right? Some information on the degree of completeness of your data would be nice in the intro. See comment above.

Line 68: Please give the full trait names before referring to the abbreviations here.

Line 69: Same as above: It might be more suitable for a broader audience if one would not have to scan through the methods to understand the abbreviations here. Would it not be better to be a bit more explicit about the traits used in your paper and then provide the abbreviations? You could do so already in the intro in line 51?

Line 74: It might help the reader if the panels would be numbered in Figure 2; in this case you could refer to the respective panel when stating the results.

Line 174-176: I wonder how much we actually know about root size in general or maximum rooting depth in particular. This is a very tricky trait to measure and rooting depth is likely underestimated (see Laughlin et al. 2023, New Phytologist). I have the feeling that this should be highlighted in the discussion and I would be more cautious when interpreting the results.

Line 182-184: Which is not necessarily increased only by being tall, but also by being big (see comment above).

Line 196-201: Interesting result!

Line 215-218: But rooting depth is strongly phylogenetically conserved (see Laughlin et al. 2023, *New Phytologist*), the same is true for the collaboration gradient (Bergmann et al. 2020, *Science Advances*). Is this in relative to the phylogenetic signal in aboveground traits? That is not clear here.

Line 236-238: When thinking about it - what about clonal extent? Clonal extend might also be an important belowground plant size trait as clonal plants can also share resources. That could be true for both woody and herbaceous plants.

Line 244-247: It is also important how these trait relationships vary across environment gradients as one would expect stronger linkages in particular environments - that could be more explicit here. I think it's also important to acknowledge the limitations of rooting depth as a trait: we still have only a very limited understanding on how deep roots can go which will also be strongly variable in different environments (you also mentioned a strong bias towards the northern hemisphere, so plants from very harsh environments are largely missing in your database). I generally like your limitations paragraph in the end of the discussion but the limitations are not only regarding the global approach, but also regarding the trait measurements itself.

Line 256: Different abbreviation for the same database? Please be consistent.

Line 256: Just out of curiosity: Would you think that an updated dataset from TRY could change your results? Would that increase the number of species in your dataset? It might help to justify why you used the same dataset as used in Carmona et al. 2021 – out of consistency and comparability, I guess?

Line 266-268: This information should be mention earlier in the introduction as this is the core dataset of your analysis.

Line 315-318: I understand that the PCA is a valuable tool for exploratory analysis of multidimensional data. Yet, PCAs not allow hypothesis testing as would be need when one aims to assess the relationship between traits in a multidimensional space (i.e., if they are orthogonal or not). You are not claiming that the eigendecomposition and hence, the comparison of angles that each pair of traits form, enable to statistically test your hypotheses, right? In that case, more caution would be required when interpreting your results.

Line 509-522: I would at least refer to the paper by Weigelt et al. 2021, especially given the deviation of the results between Carmona et al. 2021 and Weigelt et al 2021. It is not clear to me if one of these frameworks is superior over the other as the discussion seems to be ongoing - see Nature commentary of Weigelt and Carmona in 2023. This issue could also be addressed in the discussion to highlight future needs.

Line 524-533: It would help the reader if the trait labels would not that strongly overlap in the figure.

Version 1:

Reviewer comments:

Reviewer #3

(Remarks to the Author)

Dear Authors and Editor,

This is the second version of this manuscript that I have reviewed. When I reviewed an earlier version, my overall assessment was positive. I am glad that the authors carefully considered my main comments, especially regarding the debate between Carmona et al. (2021) and Weigelt et al. (2021). I also greatly appreciate the interesting insights provided by the comparison of their original database to an updated version of the TRY database. I thank the authors for their excellent work, and I have only a few minor comments remaining.

Line 141: Typo in 'Root lateral spread'

Line 173: 'Lateral root spread' instead of 'Root lateral spread' – please ensure consistent terminologies.

Manuscript Title: Aboveground and belowground sizes are aligned in the unified spectrum of plant form and function

Eleonora Beccari & Carlos P. Carmona

We are grateful for the opportunity to resubmit a revised version of our manuscript to Nature Communications. We would like to thank the editor and reviewers for their very constructive comments that have greatly aided us in improving our manuscript. Following reviewers' comments, we decided to expand the Introduction and Discussion section substantially. Specifically, we expanded the Introduction to clearly inform the readers on the controversy between Carmona et al. (2021) and Weigelt et al. (2021) as well as to provide deeper details on the ecological context of our hypothesis and on the allometric scaling hypothesis. We enhanced the Discussion to stress the possible limitations of our data, not only in terms of global coverage, but also regarding the quantification of the traits itself and the possible relation among belowground size traits and clonality-related traits. Additionally, we moved the definition of all acronyms in the Introduction, and we reduced the use of the acronym itself to the minimum possible throughout the whole text. Furthermore, we addressed the question raised regarding the inclusion of the plant height/rooting depth ratio in the analysis by testing its inclusion in the UPFS framework. Finally, we modified the title to comply with Nature Communications standards.

Below, we provide extended explanations of each point, explaining in deeper detail the reasons behind the modification of the manuscript following reviewers' recommendations (response text is in black font colour and reviewers' comments in **dark orange**). Please note that all lines mentioned in the text (e.g. Lines XX -XX) refer to the new version of the manuscript without track changes.

Reviewers comments on the initial version:

Reviewer #1 (Remarks to the Author):

The question of the dimensionality of trait space is of major importance both in terms of pure science (e.g., evolutionary and ecological constraints on morphology) and sampling design (e.g., selecting orthogonal traits to sample). The authors set forth to test a key question relating to the dimensionality of combined above-and-belowground trait spaces. They assemble a sizable dataset (given the limited availability of trait data) and take several approaches to addressing this question. The approaches used are all reasonable, and importantly, they all point to a similar conclusion: that above- and below-ground size traits form a single spectrum. The manuscript itself is in great shape and the message and writing very clear, thus I only have minor comments:

R1: Thanks for your work on our manuscript and for your support.

Figure 1 is great! It not only helps clarify some of the terminology, but also clearly sets out the main hypothesis being tested

R2: Thank you!

Line 68 -83: "Lr", "Dr", etc. - since the methods are at the end in this article format, this is the first place readers will be seeing these acronyms, so it would be good to define them here as well.

R3: We agree with the reviewer. Accordingly, we defined traits acronym in the Introduction (lines 75-80). Furthermore, we decided to remove all acronyms related to traits to better guide the reader through the text.

Work by Uyeda et al. (<https://doi.org/10.1093/sysbio/syv019>) suggests that PPCA can be misleading (at least in some contexts). Perhaps mention this somewhere in the discussion? Given that the evolutionary aspect isn't the main focus of this manuscript, and that you get similar results either way, I think simply mentioning this is fine without the need for any new analyses.

R4: We agree with this comment. PPCA is based on multivariate Brownian motion, and its results should be interpreted with caution. However, the potential issues are strongly related to the use of species scores along the PPCA dimensions rather than the identification of the main trend of traits covariation (i.e., the trait eigenvectors, Polly *et al.*, 2013). Yet, following the reviewer’s advice, we now recommend in the Discussion that PPCA results should be critically interpreted (lines 269- 273):

*“It is important to note that phylogenetically informed methods are based on specific assumptions (e.g. specific models of evolution), which require critical evaluation (Polly *et al.*, 2013; Uyeda *et al.*, 2015). Therefore, we advocate for future studies exploring phylogenetic influence in defining plant aboveground and belowground investments and the coordination among them.”*

By providing the full code and data the methods are completely transparent and reproducible. I very much appreciate the effort spent on writing and documenting the code.

Reviewer #1 (Remarks on code availability):

The code is in great shape, very well-documented and easy to read. Everything needed for reproduction appears to be there. I've read through A LOT of code, and this is among the best. I was able to run the code without any problems. Kudos to the authors!

R5: Thank you, it is very nice to read these words.

Reviewer #2 (Remarks to the Author):

The authors add rooting depth and lateral spread to a larger set of traits to ask whether root size traits alters the dimensionality of plant form and function. Determining the dimensionality of plant traits is a core objective in comparative functional ecology. The authors suggest that incorporating root size traits did not change the fundamental structure and dimensionality of the plant functional space, regardless of trait completeness and phylogenetic relatedness. Plant size emerged as a single continuum, independent from leaf and root economics strategies. While I appreciate the analysis, I have several concerns.

R6: Thanks for your comments. We hope that our reply will help in solving the concerns raised.

The authors introduce the paper as if the controversy over Carmona’s (the second author) last paper did not exist, which is difficult to overlook. Carmona *et al.* 2021 analyzed a subset of the data in this paper and concluded that leaf and root economics (nitrogen concentrations, most notably) must be independent, despite there being no mechanistic biological explanation for such an interpretation. Virtually all nitrogen in leaves must enter through the roots, so it is difficult to reconcile the model results with ecological processes. Then the authors set up two contrasting “hypotheses”, one that mentions the Weigelt *et al.* 2021 paper (which is the paper that disagreed with the Carmona *et al.* 2021 paper), but that paper also included root size, so in some ways this analysis has already been done. In sum, the authors add rooting depth to this paper to see if it changes their recent results.

R7: We have chosen to address this comment by dividing it into two parts. Each part is interconnected, but we believe each deserves a specific explanation. Accordingly, we will address (i) the controversy between Carmona *et al.* (2021) and Weigelt *et al.* (2021), including the coordination among root and leaf nitrogen; (ii) the novelty of including root size traits in a framework reconciling GSPFF and RES traits.

Regarding point (i), in the first version of the manuscript, we intentionally decided to avoid discussing the controversy between Carmona *et al.* (2021) and Weigelt *et al.* (2021), hoping to advance the science beyond the debate. The reasoning behind this choice was that the mentioned controversy had already been addressed (besides the two mentioned papers) by two peer-reviewed Matters Arising papers

(Bueno et al., 2023; Weigelt et al., 2023). However, we agree with the reviewer that informing readers about this controversy may be beneficial for defining the context behind the global synthesis of plant functional traits. Therefore, we substantially expanded the Introduction to directly address the ongoing debate (lines 43-53).

To clarify further, Weigelt et al. (2021) define a "*conservation gradient*" that involves both leaf and root tissues; this conservation gradient includes nitrogen content in the leaves and roots, but also leaf mass per area (we use its inverse, specific leaf area) and root tissue density. Conversely, the space defined by Carmona et al. (2021) depicts these four traits as part of two different dimensions (specific leaf area and leaf N content being part of the *leaf economics spectrum*, and root tissue density and root N content taking opposite directions along a *root economics axis*). Both Weigelt and Carmona use the same dataset, where the correlation between leaf nitrogen and root nitrogen content ranges between 0.27 and 0.32, depending on how the trait data is processed. Based on these correlations (and the correlations among other traits), both papers attempt to build a PCA to explore the dimensionality and structure of plant phenotypes. As shown in Bueno et al. (2023), both groups of authors reach essentially identical PCA solutions but differ in how they interpret these results. Bueno et al. (2023) demonstrate that the interpretation by Weigelt et al. (2021, 2023) is not optimal because it does not consider all relevant dimensions of variation simultaneously. It is also worth mentioning that the other two components of Weigelt et al.'s conservation gradient—leaf mass per area and root tissue density—are even more weakly correlated ($r = 0.12$). This further suggests possible issues in their interpretation of the PCA. We think that, while a priori models and hypotheses about the structure of the plant functional space are important, they should be adapted when data does not support these models, and this dataset does not show a strong correlation between leaf and root N concentration.

In any case, we think the reviewer is misinterpreting the conclusions by Carmona and Bueno et al., as these authors do not claim that "*leaf and root economics (nitrogen concentrations, most notably) must be independent.*" Instead, they argue that, according to the data currently available, these two traits are not strongly related and are closer to being orthogonal than parallel within the resulting synthetic space. This finding holds true for both the Carmona-Bueno and Weigelt solutions, as shown in Bueno et al. (2023). We think that the approach we have taken in the manuscript currently under evaluation (examining angles between pairs of traits in the resulting space and quantifying the effective dimensionality of that space and how it is modified by the addition or removal of individual traits) is a step forward towards a more nuanced and robust interpretation of the complex relationships between multiple traits.

Additionally, there are several plausible reasons why root and leaf nitrogen might not be strongly correlated when considering a large pool of species. For example, variations in nitrogen uptake mechanisms, differences in metabolic rates, and environmental influences could all contribute to the observed weak correlation. Our ongoing efforts with the TraitDivNet initiative (details here: <https://macroecology.ut.ee/en/traitdivnet/>), where the same traits are measured across many ecosystems, aim to further elucidate these questions.

We hope this expanded explanation clarifies our approach and the context of our research. We appreciate the reviewer's insights and have made the necessary revisions to address these points in our manuscript.

Referring to point (ii; the novelty of including root size traits in a framework reconciling GSPFF and RES traits), as the reviewer mentioned, we directly refer to Weigelt et al. (2021) work because is the only one that relates a single trait related to root size to a subset of traits synthesizing plant aboveground and belowground investments. After hypothesizing coordination among above- and belowground dimensions, Weigelt et al. (2021) found results contradicting their hypothesis, showing a decoupling in size investment, with rooting depth (alone) forming a single independent root size dimension.

However, more recently, the work by Tumber-Dávila *et al.* (2022) provided possibly the most extensive database of plant size traits and suggested a strong coordination among plant height, rooting depth, and lateral spread. This coordination, if included in a principal component framework, would produce results differing from those of Weigelt *et al.* (2021), but confirming their original expectations. Given these contradictory and, in some cases, unexpected results, it seemed clear to us that the matter of the inclusion of root size traits in the UPFS is still open, and our work is needed to shed light on size traits' coordination at the global scale.

We have modified our manuscript to further clarify the need for our study. In particular, we have expanded the reasoning justifying our hypotheses in the Introduction (now expanded lines 54-74) and in Figure 1. We want to clarify that, with our framework, we do not simply “*add rooting depth to this paper to see if it changes their recent results*” as the reviewer suggests. We have developed a multi-step approach aimed to test the consistency of our findings by considering separately aboveground (i.e. GSPFF from Díaz *et al.*, 2016) and belowground (i.e. RES from Bergmann *et al.*, 2020) trait subsets, considering the phylogenetic relatedness of species, and both woody and herbaceous growth forms. Thanks to this series of tests, we demonstrate that above- and belowground size traits are coordinated at the global scale, confirming Tumber-Dávila *et al.* (2022) results.

Additionally, our framework allows us to provide a possible explanation for the discrepancy we found with Weigelt *et al.* (2021). As we wrote in lines 236-250, we show how woody and herbaceous subsets have different preferential investments in size traits, which, when pooled together in a global dataset (and in proportions closer to those observed in nature, see below), define the coordination between above- and belowground dimensions underpinning our main findings. Weigelt *et al.* (2021) results are based on 540 species with common information for plant height and rooting depth, with 62.41% being woody and 33.52% herbaceous. In our work, we consider 1631 species sharing observations for the two traits, with 36.54% being woody and 58.86% herbaceous (approximately 45%-48% of global vascular plant species are estimated to be woody; FitzJohn *et al.*, 2014). Accordingly, the discrepancies between our works seem to be caused by the unbalanced proportion of growth forms as well as the reduced number of observations available for root size traits.

Thus, our work not only demonstrates the coordination between above- and belowground dimensions, confirming Tumber-Dávila *et al.* (2022) findings and reconciling with Weigelt *et al.* (2021) original hypothesis, but also suggests the importance of further increasing database extension and coverage to accurately address plant strategies. To better address this last point, we have expanded the related portion of the Discussion (lines 244-256).

The authors don't adequately address the potential relationship between aboveground height and rooting depth, which would be useful for generating more ecologically interesting hypotheses. The deepest-rooted plants on earth are also quite short in height. This is because accessing deep soil water is needed in arid climates, where it is not desirable or needed to allocate carbon to tall canopies. This is mentioned as an aside toward the very end, rather than integrated into the framing of the paper. I'm not saying that there is an inevitable trade-off globally, but allometric scaling is not the most fruitful way to approach the problem. The 'allometric scaling' hypothesis also ignores the well-known wide variation in root-to-shoot ratios that exists among plants. These ecological facts are generally ignored.

R8: Global approaches provide a synthesis of the wide diversity of strategies that plant species employ. Although in arid environments the deepest-rooted plants are generally shorter aboveground, when all worldwide habitats are simultaneously considered, these strategies are clearly less common. Therefore, they will not strongly influence the definition of global plant strategies, as our results suggest. Figures 7 and 8 in Tumber-Dávila *et al.* (2022) portray this idea quite clearly: although tree species in deserts, or generally arid climates, have, on average, the deepest roots and short height, species with other growth forms do not show a similar trend. Similarly, in humid environment trees are generally taller than deep rooted, but this trend is not maintained for other growth forms (Tumber-Dávila *et al.* 2022).

Therefore, in the search for general investments that plants make at the global scale, these cases, although important and impactful in shaping global plant strategy synthesis, are specializations to specific environmental requirements that may be not shared when plant strategies are considered across all ranges of habitats and life forms. Our results confirm this idea, showing that although rooting depth introduce higher variability in the functional spaces built considering only woody species (yet confirming the main results), this trend is not maintained in spaces built considering only herbaceous species.

Additionally, as we state in the Discussion (lines 286-288): “*In this sense, theory predicts that, after accounting for the allometric relationship among plant organs and depending on species functional type (McCarthy & Enquist, 2007; Puglielli et al., 2021), plants preferentially allocate biomass to the organ facing the greatest resource limitation at the expense of other organs (Schenk & Jackson, 2002; McCarthy & Enquist, 2007; Smith-Martin et al., 2020)*”. Thus, to maintain an extensive root apparatus (both in its depth or lateral spread), plants require an aboveground apparatus capable of sustaining such roots on a mechanical and physiological level. Similarly, to sustain extensive aboveground dimensions, roots need to provide mechanical and physiological support needed to provide aboveground organs. After accounting for these fundamental allometric constraints, plants may then allocate biomass depending on the limiting resource and the environment they live, according to the functional strategy they employ. Therefore, justifying the coordination we found among above- and belowground dimensions through the allometric hypothesis, we are not ignoring variation in root-shoot investments. Rather, we are stating that this variation exists, but the greatest portion of global plant size variation it is first accounted by the coordinated investment is size that plants should do to address these physical constraints. Below, in reply R9, we directly prove this concept by incorporating the ratio between plant height and rooting depth into our UPFS framework. This analysis (results in Table R1 and R2) allows to demonstrate that even when including this ratio in UPFS framework, which indicates the deviation from a perfect coordination between plant height and rooting depth, results remain consistent with those discussed in the main analysis.

It is odd that the authors only use the raw maximum depth and maximum spread, rather than incorporating the ratios or relationships, given that these were important in the New Phytologist paper that introduced the RSIP database. Could the authors see if these dimensions of rooting architecture interact with the other traits?

R9: We decided to include maximum rooting depth (and maximum lateral root spread) for three main reasons: to replicate a framework comparable with that of Weigelt et al. (2021), to avoid mathematical complications while simultaneously enhancing the interpretation of the trait space, and because it is known to be strongly related to life form.

1. A framework comparable to Weigel et al: Our work aims at exploring whether the inclusion of root size information alters the UPFS dimensionality. As discussed in R7, despite considering one third of the number of species compared to our dataset, Weigelt et al. (2021) was the first attempt to include root size in a framework unifying above- and belowground plant traits, thus it defines a fundamental baseline for defining our hypothesis regarding the impact of extensive root size information on the UPFS. Weigelt et al. (2021) considered maximum rooting depth as root size information, therefore one reason for the inclusion of this trait was to allow for comparison of our results with this baseline work, which define the alternative hypothesis of our work.
2. Mathematical complications: It is crucial to consider the type of information provided by a ratio. A ratio between two variables indicates the extent of their deviation from a perfect relationship. Therefore, the more strongly the two original variables are correlated, the lower the amount of information contained in the ratio. The problem is that when the ratio is later used as one of the variables to make a PCA it will get artificially inflated in importance after scaling. For an extreme example of this, imagine that root and shoot are almost perfectly correlated ($r = 0.99$); in such a

case, the ratio between the two variables is biologically meaningless, since it will be almost constant across all species. However, to introduce the ratio in a PCA we would need to scale it (make it zero mean and unit variance), which would distort the PCA results, complicate the interpretation and overemphasize patterns (small deviations from that constant) that are not biologically relevant. Indeed, the correlations between plant height and both rooting depth and lateral spread are not as strong as in this example, but they are relatively high (see below Table R1), so that the variability in ratio values will be small and these problems would arise if included in the PCA.

3. There are a priori reasons to expect that the ratio between above- and belowground size is strongly related to life form, since woody species exhibit much larger deviations from the 1:1 line than herbaceous species (as shown in Tumber-Dávila *et al.* 2022). If so, the ratio would not add a new dimension of variation, but rather be aligned with the size dimension of the UPFS, which separates woody and herbaceous species.

To show these points, we calculated the ratio between plant height and rooting depth, which was then log-transformed. We explored the correlations among UPFS traits and the ratio (Table R1), and we examined whether the inclusion of this ratio in the space built using the UPFS set of traits (excluding plant height and both root size traits) modifies our findings (Table R2). We selected the plant height and rooting depth ratio simply because these two traits showed the lowest correlation coefficient ($\rho = 0.54$) compared to plant height and lateral spread ($\rho = 0.71$), and thus their ratio may potentially retain a higher proportion of variation not captured by the individual traits.

The ratio was strongly correlated with traits defining the aboveground size axis (*sensu* Díaz *et al.*, 2016; Table R1). Accordingly, the inclusion of the ratio in the UPFS does not modify the trait structure defining the trait space observed in our main analysis (Table R2), with the ratio loading on the first dimension together with specific stem density (ssd) and seed mass (sm), and showing a low angle (thus strong positive correlation in the multidimensional trait space) with ssd and sm. Indeed, if we correlate trait angles of the space built considering only the plant height/rooting depth ratio with the one shown in the main text built considering both plant height and rooting depth separately, Pearson's correlation is equal to 0.988 (p -value < 0.001). This analysis confirms what is stated above: in the case of correlated traits, the inclusion of the ratio of two traits may potentially complicate the interpretation of the results, and in the case of a principal component analysis, it may introduce unwanted issues related to scaling and linearity of the variables.

Because of all these reasons, we have decided not to include the ratio in the PCA that we present in the manuscript.

Table R1: Pearson's Correlation Coefficients among pairs of traits defining the UPFS and size ratio. All traits have the same acronym as in the main text. Specifically, **LogRatio** is the log transformed ratio among plant height and maximum rooting depth. In the upper triangle are shown the *p*-values for each pairwise correlation (yellow color), while in the lower triangle is showed **Pearson's ρ** for each pairwise correlation (salmon pink color).

Traits Correlations	la	ln	ph	sla	ssd	sm	SRL	D	RTD	RN	Dr	Lr	LogRatio	Pearson's ρ -values	
la		0.00	0.00	0.00	0.00	0.00	0.00	0.01	0.15	0.89	0.00	0.00	0.00		
ln	0.17		0.00	0.00	0.00	0.18	0.78	0.86	0.00	0.00	0.01	0.02	0.00		
ph	0.57	-0.09		0.00	0.00	0.00	0.03	0.75	0.12	0.03	0.00	0.00	0.00		
sla	0.16	0.56	-0.21		0.00	0.00	0.00	0.04	0.00	0.00	0.00	0.00	0.01		
ssd	0.27	-0.23	0.75	-0.53		0.00	0.24	0.00	0.00	0.00	0.00	0.00	0.00		
sm	0.40	-0.02	0.61	-0.24	0.57		0.00	0.00	0.06	0.62	0.00	0.00	0.00		
SRL	-0.09	0.01	-0.06	0.18	-0.04	-0.24		0.00	0.00	0.11	0.00	0.00	0.83		
D	0.09	-0.01	-0.01	-0.07	-0.13	0.16	-0.78		0.00	0.00	0.00	0.17	0.46		
RTD	-0.05	-0.12	0.05	-0.13	0.21	0.06	-0.15	-0.30		0.00	0.08	0.64	0.97		
RN	0.00	0.32	-0.07	0.14	-0.13	0.02	-0.05	0.14	-0.25		0.08	0.02	0.02		
Dr	0.21	-0.07	0.54	-0.31	0.52	0.44	-0.25	0.14	0.09	-0.10		0.00	0.00		
Lr	0.23	-0.08	0.71	-0.27	0.64	0.49	-0.15	0.07	0.03	-0.15	0.59		0.00		
LogRatio	0.23	-0.17	0.77	-0.07	0.50	0.33	0.01	-0.04	0.00	-0.13	-0.12	0.39			
Pearson's ρ															

Table R2: Eigendecompositions of the UPFS traits subset excluding plant height and roots size traits but including plant height/rooting depth ratio (**LogRatio**). **Cum. variance**, cumulative variance explained by each axis; **N. of dimensions**, number of relevant dimensions retained; **Angles**, angles across pairs of traits considering all relevant dimensions; **END**, effective number of dimensions.

UPFS and ph/Dr											
	PC1	PC2	PC3	PC4	PC5	PC6	PC7	PC8	PC9	PC10	
Eigenvalues	2.55	1.96	1.63	1.08	0.92	0.68	0.50	0.40	0.18	0.10	
Cum. variance	25.50	45.07	61.34	72.17	81.39	88.21	93.21	97.16	98.98	100.00	
Loadings	PC1	PC2	PC3	PC4	PC5	PC6	PC7	PC8	PC9	PC10	
la	0.30	0.34	-0.62	0.18	0.22	-0.42	0.36	0.13	0.06	0.00	
ln	-0.49	0.35	-0.53	0.26	-0.22	0.08	-0.38	0.27	0.10	-0.04	
sla	-0.66	0.16	-0.49	0.26	0.29	0.20	0.04	-0.22	-0.24	0.04	
ssd	0.85	-0.12	-0.24	-0.09	-0.22	0.01	-0.11	0.23	-0.29	0.01	
sm	0.69	0.33	-0.34	0.03	-0.19	-0.13	-0.25	-0.44	0.06	0.00	
SRL	-0.29	-0.74	-0.44	-0.33	0.00	-0.14	-0.02	-0.08	-0.01	-0.21	
D	0.10	0.85	0.40	-0.06	0.21	0.04	-0.02	0.01	-0.08	-0.21	
RTD	0.28	-0.34	0.13	0.79	-0.29	0.18	0.18	-0.05	0.01	-0.11	
RN	-0.30	0.42	-0.19	-0.35	-0.65	0.19	0.33	-0.05	-0.01	0.00	
LogRatio	0.57	-0.04	-0.38	-0.20	0.33	0.60	0.07	0.04	0.13	-0.02	
Angles	la	ln	sla	ssd	sm	SRL	D	RTD	RN	LogRatio	
la		0.00	58.68	72.84	60.31	35.71	99.68	85.03	87.71	78.25	50.17
ln	58.68		0.00	17.34	117.87	92.88	88.21	88.04	98.69	56.72	103.54
sla	72.84	17.34		0.00	128.08	108.25	76.30	99.68	96.51	62.48	113.08
ssd	60.31	117.87	128.08		0.00	33.07	91.54	97.36	77.55	113.22	20.82
sm	35.71	92.88	108.25	33.07		0.00	111.76	74.75	85.77	91.80	32.18
SRL	99.68	88.21	76.30	91.54	111.76		0.00	152.75	99.54	91.98	81.89
D	85.03	88.04	99.68	97.36	74.75	152.75		0.00	107.43	63.87	100.16
RTD	87.71	98.69	96.51	77.55	85.77	99.54	107.43		0.00	152.64	92.69
RN	78.25	56.72	62.48	113.22	91.80	91.98	63.87	152.64		0.00	95.96
LogRatio	50.17	103.54	113.08	20.82	32.18	81.89	100.16	92.69	95.96		0.00
N. of dimensions	4										
END	6.28										

The paper has many unhelpful acronyms that made the paper very difficult to read: GSPFF, RES, UPFS, END, Dr, Lr, etc... It is especially not helpful to abbreviate rooting depth (Dr) and lateral spread (Lr) by using the name of these traits in the RSIP database. Just spell them out. Moreover, they aren't even defined at first mention on L 68. END is not defined at first mention L 85. These choices undermine the readability of the ms.

R10: We moved the explanation of the traits' acronym to the beginning of the text (lines 75-80) and defined END at its first mention (line 90). Throughout the text, we tried to use the full names of the traits as frequently as possible, drastically reducing the use of acronyms for functional traits.

Reviewer #3 (Remarks to the Author):

Dear authors and Editor,

I have now read the review manuscript entitled "Above and belowground size form a single dimension in the unified spectrum of plant form and function". In this paper the authors studied how traits related to plant size, particularly of belowground plant organs (i.e., rooting depth and root lateral spread) fit into the recently established unified spectrum of plant form and function. In their study, the authors test two main contrasting hypotheses regarding the coordination of above- and belowground size: First, that there is only a weak relationship between above- and belowground sizes, as in contrast to plant height, root size may not solely explain resource acquisition belowground. Alternatively, allometric scaling results in strong relationships in biomass partitioning across plant organs (if you are big above, then also belowground), which results in a strong relationship between above- and belowground sizes. The authors present interesting findings which support the latter, demonstrating that both above- and belowground plant sizes are strongly aligned and forming a single dimension of variation. This confirms previous evidence, but now also on a whole plant perspective including both above- and belowground traits of the global spectrum of plant form and function and the root economic space. I very much enjoyed reading the manuscript and the work will be of great interest in the current debate about the dimensionality of above- and belowground form and function; the results are very well presented and the data is rigorously analysed. However, while I like the limitations paragraph in the discussion it would be better to also address potential limitations of the data, not only in terms of global coverage, but also regarding the quantification of the traits itself. The database is certainly impressive but we still have lots to do in order to better understand how deep plants can root, and why. I also felt that that the claim of an independent above- and belowground dimension of the authors could be more nuanced, given the still ongoing debate which was issued in a commentary in Nature. Further, I think some further clarification could help to the reader to better digest the methods (which are quite complex). Nevertheless, this is a thorough, well-written contribution, and I enjoyed reading the manuscript. See below my more detailed comments that might be taken into consideration:

R11: Thank you for your appreciation and comments. We have carefully addressed each of them. Specifically, we have expanded the Introduction to provide further context to the debate about the dimensionality of trait spaces, addressing both above- and belowground plant investments (lines 29-53). Additionally, we have substantially expanded the Discussion to address the issues raised by the reviewer in their point-by-point comments, particularly considering the limitations of our approach in relation to our findings (lines 215-314).

Comments:

Line 35-37: You state that the unified plant functional space demonstrated the independence of above- and belowground economics at a global scale. Given the current debate about the independence of above- and belowground dimensions, which is still ongoing until now, I would advocate for a more

nuanced view here. At least refer to the contrasting findings by Weigelt et al. 2021 (New Phytologist) and/or the commentary in Nature (Weigelt et al. 2023).

R12: We agree that directly addressing these two contrasting findings is important, particularly for readers that are not aware of the state of art of debate about the dimensionality of trait spaces addressing above- and belowground plant investments. Accordingly, we expanded the relative section in the Introduction highlighting the ongoing debate and specifying the differences between the two studies as follows (lines 43-53):

“The separated spaces considering aboveground and fine root traits further highlighted the need of an integrated global synthesis of plants’ above- and belowground components (Laughlin, 2014, 2023). Carmona et al. (2021) combined the traits defining the GSPFF and RES and defined a unified plant functional space (UPFS) demonstrating that four dimensions are needed to portray global plant aboveground and fine-roots strategies (Fig. 1). The main features of the UPFS reflect those of the individual GSPFF and RES planes, revealing the independence of above- and belowground economics at a global scale (Carmona et al., 2021) (Fig. 1). In contrast, Weigelt et al. (2021) proposed a potential coordination between the leaf economics spectrum and the conservation dimension of the RES (see also Weigelt et al., 2023). However, Bueno et al. (2023) supported the integrity of the UPFS, suggesting that the divergent conclusions among the two studies stems from different interpretations of trait alignment in otherwise equivalent spaces.”

Line 41-44: It could also be possible that adult plant height is not a good proxy for resource investment aboveground, particularly in herbaceous plants which strongly vary in morphology (e.g. rosette, tillers). Shoot biomass could be better suited to characterize the plant size gradient at least for herbaceous plants; see Funk et al. 2024, Functional Ecology.

R13: We agree that, particularly in some specific herbaceous cases, plant height may not be the best proxy for aboveground investment. However, plant height is one of the most widely used and available traits for plants (in our dataset we have measurement for plant height for 13,438 species), thus providing a good compromise to address plant aboveground investments at the global scale. However, we agree that the incorporation of shoot mass could provide interesting insights particularly for herbaceous plants. Therefore, we addressed this point in the Discussion (lines 224-228):

“However, it is important to acknowledge that despite the high accessibility and prevalent use of plant height as a proxy for aboveground plant biomass (Wright et al., 2004; Reich, 2014; Díaz et al., 2016b; Weigelt et al., 2021; Carmona et al., 2021), plant height may not be the optimal proxy for herbaceous species (Funk et al., 2024), particularly those with morphologies that do not detach significantly from the ground (e.g., rosettes or tillers). In such cases, shoot biomass is suggested to provide a more accurate representation of aboveground investments (Funk et al., 2024).”

Line 52-54: What does this number tell me in this context? I doubt that you have complete trait information for even a small fraction of these species so this number sounds like an overstatement here. I would at least expect some more detailed information about the size of complete cases when you include belowground plant size traits.

R14: We used a dataset of 39,334 species and twelve traits. Each trait has a different level of completeness, and the completeness information for each trait is specified in Supplementary Table 1a. Each species has at least one measurement for each trait. The trait with the most complete information is seed mass, with 23,003 species having seed mass measurements, whereas the least complete is root nitrogen concentration, with 1,230 species having measurements. Depending on the subset of traits considered, the dataset will have different numbers of species with complete trait data. For instance, the UPFS trait subset has 264 species with complete information for all considered traits, while the UPFS trait subset including both Lr and Dr has 134 species with complete information for all traits (Supplementary Table 1b). Although the number of species with complete information for all traits is

low, pairwise trait correlations will have varying levels of completeness, and species identity will vary within the set of 39,334 species present in the dataset. For instance, plant height and seed mass have 7,217 species with observations for both traits, while plant height and root nitrogen have 869 species. Accordingly, we believe it is still important to mention the full extent of the dataset, given that each pairwise trait correlation considered in defining the UPFS involves different species subsets.

However, we agree with the reviewer that at least some general information on the completeness of the dataset should be included in the main text to avoid misleading readers about the database completeness. Therefore, we have expanded the Introduction to include this point, as reported below (lines 80-85). We would like to highlight that information about the completeness of individual traits in the dataset is reported in Supplementary Table 1a, while information about completeness among pairwise trait correlations is found in Supplementary Fig. 2. Information about completeness among the different sets of traits considered (UPFS; UPFS + Lr; UPFS + Dr; UPFS + both root traits; and equivalent trait subsets for GSPFF and RES) is provided in Supplementary Table 1b.

“Accordingly, we assemble a dataset encompassing 39,334 plant species with different levels of traits completeness (Supplementary Fig. 1), ranging from a maximum of 23,003 species with data for seed mass to a minimum of 1,230 species for root nitrogen concentration (Supplementary Table 1a). 134 species had complete observations for all twelve traits, whereas 2,965 plant species had observations for at least one root size trait (Supplementary Table 1b).”

Line 64-65: That might have a huge effect on your analyses, right? Some information on the degree of completeness of your data would be nice in the intro. See comment above.

R15: The set of analyses repeated on the subset of traits with complete trait information, indeed, is devoted to test the consistency of our approach. We included information on trait completeness in the new version of the Introduction (lines 80-85), please see previous comment R14 for details.

Line 68: Please give the full trait names before referring to the abbreviations here.

R16: We expanded the Introduction section in order to define traits acronym earlier in the text (lines 76-80). Furthermore, we drastically reduced the use of acronyms across the text.

Line 69: Same as above: It might be more suitable for a broader audience if one would not have to scan through the methods to understand the abbreviations here. Would it not be better to be a bit more explicit about the traits used in your paper and then provide the abbreviations? You could do so already in the intro in line 51?

R17: Done, see comment R16 and Introduction (lines 76-80).

Line 74: It might help the reader if the panels would be numbered in Figure 2; in this case you could refer to the respective panel when stating the results.

R18: Done, Figure 2 contains panels, and the text is updated.

Line 174-176: I wonder how much we actually know about root size in general or maximum rooting depth in particular. This is a very tricky trait to measure and rooting depth is likely underestimated (see Laughlin et al. 2023, New Phytologist). I have the feeling that this should be highlighted in the discussion and I would be more cautious when interpreting the results.

R19: We agree. Rooting depth is a trait of difficult measurement and generally underestimated. We expanded the Discussion to highlight this possible limitation inherent of trait measurements as follows (lines 308 -311):

“Furthermore, it is important to emphasize the inherent limitations of trait measurements. Traits that are difficult to measure may present biases in their estimation. For instance, our limited understanding

of root size (Tumber-Dávila et al., 2022) is primarily due to the challenges in extracting measurements for the deepest-rooted species, leading to a general underestimation of global root size (Laughlin et al., 2023)."

Line 182-184: Which is not necessarily increased only by being tall, but also by being big (see comment above).

R20: As stated in reply R13, the Discussion now directly addresses plant size as a proxy of aboveground investments and its limitation particularly in relation to herbaceous species (lines 224-228):

"However, it is important to acknowledge that despite the high accessibility and prevalent use of plant height as a proxy for aboveground plant biomass (Wright et al., 2004; Reich, 2014; Díaz et al., 2016b; Weigelt et al., 2021; Carmona et al., 2021), plant height may not be the optimal proxy for herbaceous species (Funk et al., 2024), particularly those with morphologies that do not detach significantly from the ground (e.g., rosettes or tillers). In such cases, shoot biomass is suggested to provide a more accurate representation of aboveground investments (Funk et al., 2024)."

Line 196-201: Interesting result!

R21: Thank you!

Line 215-218: But rooting depth is strongly phylogenetically conserved (see Laughlin et al. 2023, *New Phytologist*), the same is true for the collaboration gradient (Bergmann et al. 2020, *Science Advances*). Is this in relative to the phylogenetic signal in aboveground traits? That is not clear here.

R22: RES dimensions exhibit comparable dimensionality (i.e., effective number of dimensions) between phylogenetically informed and non-phylogenetically informed spaces, whereas GSPFF dimensions display higher dimensionality in the phylogenetically informed space compared to the non-phylogenetically informed space. Therefore, the dimensionality of belowground traits is nearly identical to that observed in phylogenetically independent axes. This does not imply that collaboration or conservation gradients are not phylogenetically conserved. Instead, it suggests that the dimensions defining RES are phylogenetically independent. In contrast, the dimensionality required to depict aboveground traits increases in the phylogenetically informed space, suggesting that along GSPFF dimensions, species positions still retain some unexplained phylogenetic dependence. Thus, species investments across GSPFF dimensions may still be at least partially influenced by their phylogenetic history.

Below, we estimated the phylogenetic signal (expressed as Pagel's Lambda) along the main dimensions of trait variation constructed considering UPFS, GSPFF, and RES trait subsets with complete information (Table R3). On average, all dimensions exhibit high phylogenetic signal, confirming our previous statement: within each dimension, species phylogenetic signal may still be high. However, when considering dimensions that are phylogenetically independent of one another (i.e., phyl-PCA axes, Revell, 2009; Polly et al., 2013; Revell & Harmon, 2022), phylogenetically informed RES (Supplementary Table 14) exhibit comparable dimensionality and structure to non-phylogenetically informed RES (Supplementary Table 6).

We agree that the sentence as originally written does not clearly convey this message to the readers. Therefore, we have revised the sentence as shown below (lines 262-273). Furthermore, we emphasized the need for cautious interpretation of results derived from phylogenetically informed PCA.

"Conversely, the higher dimensionality found in phylogenetically informed spaces built on GSPFF traits suggests a phylogenetic coordination among aboveground dimensions (Fig. 1), with the positioning of species along plant size (considering both above- and belowground sizes) and leaf economics dimensions appearing to be more closely tied by the evolutionary history of species (Capdevila et al., 2023) (Supplementary Fig. 6). Accordingly, the evolution of above- and belowground

global plant strategies may have been subject to different selection pressures (Carmona et al., 2021) (e.g., differences in resource distribution among above and belowground components (Laughlin, 2023), potentially leading to a decoupled evolutionary trajectory in belowground but not in aboveground investments. It is important to note that phylogenetically informed methods are based on specific assumptions (e.g. specific models of evolution), which require critical evaluation (Polly et al., 2013; Uyeda et al., 2015). Therefore, we advocate for future studies exploring phylogenetic influence in defining plant aboveground and belowground investments and the coordination among them”

Table R3: Pagel's lambda calculated for each dimension defining UPFS, GSPFF, and RES traits subset. We considered trait dimensions of the PCA analyses performed on the set of traits with complete records only (Supplementary Table 1b).

Pagel's lambda		Pagel's lambda		Pagel's lambda	
UPFS + Root size		GSPFF + Root size		RES + Root size	
Plant size	0.97	Plant size	0.96	Root size	0.92
Leaf economics	0.88	Leaf economics	0.87	Root Collaboration	0.86
Root Collaboration	0.84			Root Conservation	0.79
Root Conservation	0.86				
UPFS		GSPFF		RES	
Plant size	0.97	Plant size	0.97	Root Collaboration	0.82
Leaf economics	0.85	Leaf economics	0.87	Root Conservation	0.61
Root Collaboration	0.94				
Root Conservation	0.76				

Line 236-238: When thinking about it - what about clonal extent? Clonal extend might also be an important belowground plant size trait as clonal plants can also share resources. That could be true for both woody and herbaceous plants.

R23: We believe that clonality-related traits will be the future step forward in the expansion of UPFS framework. Accordingly, we agree with the reviewer that clonality-related traits should be mentioned in the Discussion. We included it in lines 291 – 297 as follows:

“Similarly, this difference in variability among root size traits may stem from covariation with other plant investments not included in the framework. Indeed, clonality-related traits may play a role in influencing the complexity of above- and belowground size traits coordination, particularly in UPFS framework (Klimešová et al., 2021; Laughlin, 2023; Chelli et al., 2024). For instance, clonal herbaceous traits, (e.g. lateral spread of clonal organs) positively correlates with traits related to aboveground size (Klimešová et al., 2016; Chelli et al., 2024), suggesting a possible role of clonality in influencing the covariation among above- and belowground biomass investments.”

Line 244-247: It is also important how these trait relationships vary across environment gradients as one would expect stronger linkages in particular environments - that could be more explicit here. I think it's also important to acknowledge the limitations of rooting depth as a trait: we still have only a very limited understanding on how deep roots can go which will also be strongly variable in different environments (you also mentioned a strong bias towards the northern hemisphere, so plants from very harsh environments are largely missing in your database). I generally like your limitations paragraph in the end of the discussion but the limitations are not only regarding the global approach, but also regarding the trait measurements itself.

R24: We agree with the reviewer that limitations of our work can be expanded. Therefore, we expanded the section “*Rooting depth and the limits of global traits-based approach*” to inform readers about the potential bias associated with traits that are difficult to measure (as discussed in R16 above). We expanded the Discussion as follows (lines 308-314):

“Furthermore, it is important to emphasize the inherent limitations of trait measurements. Traits that are difficult to measure may present biases in their estimation. For instance, our limited understanding of root size (Tumber-Dávila et al., 2022) is primarily due to the challenges in extracting measurements for the deepest-rooted species, leading to a general underestimation of global root size (Laughlin et al., 2023). Nevertheless, with the most up-to-date datasets, our global analysis indicates a strong, overarching alignment of above- and belowground size traits that is independent of growth form and phylogeny. This suggests a unified continuum of size-related traits across the plant kingdom, distinct and independent from leaf and root economic strategies.”

Line 256: Different abbreviation for the same database? Please be consistent.

R25: Amended

Line 256: Just out of curiosity: Would you think that an updated dataset from TRY could change your results? Would that increase the number of species in your dataset? It might help to justify why you used the same dataset as used in Carmona et al. 2021 – out of consistency and comparability, I guess?

R26: We decided to use the same dataset as Carmona *et al.* (2021) to ensure comparability and consistency with previous work. To address your curiosity, we downloaded the updated TRY database (i.e., TRY version 6; Kattge *et al.*, 2020) to briefly explore the differences in trait completeness between our dataset and the TRY version 6 dataset (Table R4 below). Then, we used the TRY6 data to explore trait correlations in this updated version, which we compared with those in our dataset (Table R5). It is important to note that we processed the trait information relatively quickly, and the number of species present for each trait in the TRY version 6 (TRY6) database may slightly decrease after more detailed data processing. Similarly, fine root traits in the TRY6 dataset have not been corrected for the study design, unlike in our analyses, where fine root data showed this correction (Carmona *et al.* 2021; Bergmann *et al.* 2020). Thus, fine-root correlation coefficients in TRY6 may have process-related bias. Correction procedures require significant effort and time in data cleaning and processing; therefore, we decided to skip this step for this exploration and provide rough results. However, if reviewers consider this information important for the manuscript, we will address the issue further in detail.

As shown in Table R4, TRY6 significantly increases the number of observations for aboveground traits. Nevertheless, aboveground traits are extensively documented and have had good global coverage since seminal global datasets (e.g., GSPFF ones, Díaz *et al.*, 2016). Accordingly, although the enhanced aboveground information in the TRY6 database is substantial, the results regarding the coordination between above- and belowground traits would not change if TRY6 information was used, given that the dataset used by Carmona *et al.* (2021) already accounts for the broad variation of aboveground forms and functions across the globe. Accordingly, trait correlations within TRY6 were consistent with those observed in our dataset (Table R5).

Focusing on root traits, our dataset generally has a higher number of observations for belowground traits compared to TRY6. Root traits are generally underrepresented in global trait databases (Iversen *et al.*, 2017; Bergmann *et al.*, 2020; Carmona *et al.* 2021), and future sampling efforts aimed at expanding the coverage of these traits may potentially provide further insights into how root traits fit into the UPFS. However, the higher coverage we have for root traits compared to TRY6 allows us to confidently state that we present the most up-to-date version of the UPFS available. Fine-root trait correlations within TRY6 were generally consistent with those observed in our dataset (Table R5), yet the few inconsistencies present (i.e., correlation among ph-N and Dr-N) may be related to the different processing of fine root data mentioned above. Importantly, lateral root spread was the only trait showing

inconsistent correlations compared to our database (Table R5). However, observations for lateral root spread in the TRY6 dataset are 32 compared to the 1,757 observations for the same trait in our dataset. Thus, we believe that TRY6 correlation coefficients are insufficient to adequately describe global variation in root lateral spread compared to our dataset.

We are confident that these results demonstrate that the inclusion of TRY6 root data will not significantly modify our findings. Indeed, correlation coefficients of TRY6 are significantly correlated with those in our work ($\rho = 0.79$, p -value < 0.001), suggesting consistent correlations between the two trait datasets.

Table R4: Comparison among the dataset we used and the new version 6 of TRY database. *N. species*, shows the number of species with observations for each trait; *N. Species TRY6*, shows the number of species with observations for each trait in the version 6 of TRY database.

Trait	N. Species	N. Species TRY6
Plant height	13438	26776
Specific stem density	11148	13719
Seed mass	23003	39193
Leaf area	12766	16374
Leaf nitrogen concentration	9095	11380
Specific leaf area	9601	17102
Specific root length	1697	828
Root diameter	1550	626
Root tissue density	1361	411
Root nitrogen concentration	1230	449
Maximum rooting depth	2598	2985
Maximum lateral root spread	1757	32

Table R5: Pearson's Correlation Coefficients among pairs of traits defining the UPFS are shown for both the dataset used in the main analyses (Beccari & Carmona, upper triangle, light orange) and the dataset from TRY 6 (lower triangle, light green). All traits have the same acronyms as in the main text, and all traits are log-transformed and scaled. Please note that the fine root traits from TRY 6 have not been corrected for study design, unlike the data from Beccari & Carmona.

		Beccari & Carmona											
		la	ln	ph	sla	ssd	sm	SRL	D	RTD	RN	Dr	Lr
TRY 6	la		0.17	0.57	0.16	0.27	0.40	-0.09	0.09	-0.05	-0.01	0.21	0.23
	ln	0.16		-0.09	0.56	-0.23	-0.02	0.01	-0.04	-0.12	0.32	-0.07	-0.08
	ph	0.57	-0.07		-0.21	0.75	0.61	-0.06	0.00	0.05	-0.04	0.54	0.71
	sla	0.17	0.52	-0.21		-0.53	-0.24	0.18	-0.07	-0.13	0.13	-0.31	-0.27
	ssd	0.25	-0.24	0.69	-0.51		0.57	-0.04	-0.12	0.21	-0.11	0.52	0.64
	sm	0.42	0.00	0.61	-0.20	0.51		-0.24	0.15	0.06	0.04	0.44	0.49
	SRL	-0.14	-0.03	-0.28	0.21	-0.23	-0.21		-0.78	-0.15	-0.03	-0.25	-0.15
	D	0.13	-0.06	0.20	-0.16	0.17	0.20	-0.64		-0.30	0.08	0.14	0.07
	RTD	0.00	-0.18	0.21	-0.20	0.41	0.13	-0.16	-0.31		-0.28	0.09	0.03
	RN	0.30	0.31	0.46	0.16	0.18	0.47	-0.03	0.16	-0.04		-0.07	-0.10
	Dr	0.20	-0.07	0.55	-0.24	0.43	0.44	-0.15	0.18	0.13	0.40		0.59
	Lr	0.20	-0.26	0.92	-0.31	0.77	0.55	-0.16	0.75	-0.77	0.60	0.13	

Line 266-268: This information should be mention earlier in the introduction as this is the core dataset of your analysis.

R27: The core dataset of our work is the combination of the database extracted from Carmona et al. (2021) with the one provided in Tumber-Dávila et al. (2022). However, we agree that information on the completeness of root size information should be mentioned earlier in the Introduction. Accordingly, we expanded the introduction as specified in R14 and as reported below (lines 80-85). Furthermore, we reorganized the relative part of the Methods to let the reader better understand the how our database was compiled (lines 326-352).

“Accordingly, we assemble a dataset encompassing 39,334 plant species with different levels of traits completeness (Supplementary Fig. 1), ranging from a maximum of 23,003 species with data for seed mass to a minimum of 1,230 species for root nitrogen concentration (Supplementary Table 1a). 134 species had complete observations for all twelve traits, whereas 2,965 plant species had observations for at least one root size trait (Supplementary Table 1b).”

Line 315-318: I understand that the PCA is a valuable tool for exploratory analysis of multidimensional data. Yet, PCAs not allow hypothesis testing as would be need when one aims to assess the relationship between traits in a multidimensional space (i.e., if they are orthogonal or not). You are not claiming that the eigendecomposition and hence, the comparison of angles that each pair of traits form, enable to statistically test your hypotheses, right? In that case, more caution would be required when interpreting your results.

R28: We have used PCA primarily as an exploratory tool to summarize and visualize the multidimensional variation in plant traits. Our aim was to examine whether the inclusion of root size traits would alter the dimensionality and structure of trait correlations defining plant functional strategies. We agree with the reviewer that PCA, used as it is, does not provide means for formal hypothesis testing regarding the orthogonality of traits or the statistical significance of their relationships.

We would like to clarify that our eigendecomposition is meant to extract the main axes of independent trait variation needed to summarize global plant investments. The extraction of angles between trait pairs was intended to provide a visual and intuitive understanding of trait correlations within the main axes of the multidimensional space and to graphically represent them. This is stated in lines 376-384:

“For each eigendecomposition, we estimated trait loadings, which reflect the correlation between traits and relevant dimension of trait variation (Legendre & Legendre, 2012). Then, to understand the relationships among traits in each reduced space, we estimated the correlations between their loadings considering all relevant dimensions in the space (Supplementary Fig. 1). Following Bueno et al. (2023), we expressed these correlations as the angles that each pair of traits form in the reduced space; this procedure allows graphically representing the relationships between traits, easing interpretations of trait-trait relationships in the reduced space (Carmona et al., 2021; Bueno et al., 2023) (Supplementary Table 2). Finally, we compared the dimensionality and trait correlations in each of the four spaces built progressively including root size traits (Supplementary Fig. 1).”

After dimensionality reduction (i.e. discarding PCs that capture little variation), and when the resulting synthetic space has high dimensionality (dimensionality > 2), interpreting trait relationships defining space’s dimensions through projections of eigenvectors along single dimensions (e.g., loadings) or two-dimensional planes can lead to suboptimal interpretations of these relationships (see Bueno et al. 2023). Instead, we propose to estimate the angles among all traits’ loadings in the reduced space (or correlations, but we think angles are more intuitive for most people than correlations) to better understand trait relationships in the multidimensional space. Accordingly, we use this tool to explore trait correlations defining the main axes of trait variation, not to statistically test these relationships.

Line 509-522: I would at least refer to the paper by Weigelt et al. 2021, especially given the deviation of the results between Carmona et al. 2021 and Weigelt et al. 2021. It is not clear to me if one of these frameworks is superior over the other as the discussion seems to be ongoing - see Nature commentary of Weigelt and Carmona in 2023. This issue could also be addressed in the discussion to highlight future needs.

R29: We agree that this debate was not sufficiently explained. As explained in previous responses to Reviewer 1 queries (see R7) and in R12, we decided to include details about the ongoing debate in the Introduction as follows (lines 43-53):

“The separated spaces considering aboveground and fine root traits further highlighted the need of an integrated global synthesis of plants’ above- and belowground components (Laughlin, 2014, 2023). Carmona et al. (2021) combined the traits defining the GSPFF and RES and defined a unified plant functional space (UPFS) demonstrating that four dimensions are needed to portray global plant aboveground and fine-roots strategies (Fig. 1). The main features of the UPFS reflect those of the individual GSPFF and RES planes, revealing the independence of above- and belowground economics at a global scale (Carmona et al., 2021) (Fig. 1). In contrast, Weigelt et al. (2021) proposed a potential coordination between the leaf economics spectrum and the conservation dimension of the RES (see also Weigelt et al., 2023). However, Bueno et al. (2023) supported the integrity of the UPFS, suggesting that the divergent conclusions among the two studies stems from different interpretations of trait alignment in otherwise equivalent spaces.”

Given that the differences between the two studies are already presented in the Introduction, we prefer to avoid further discussing the reasons behind the mismatch in findings in the Discussion. Nevertheless, we would like to point out that, in our opinion, the discussion is not ongoing, unless new data arises to bring light into these relationships; we are working hard to collect such new data through the TraitDivNet initiative (details here: <https://macroecology.ut.ee/en/traitdivnet/>). Bueno et al. (2023) provided convincing responses to all the concerns suggested by Weigelt and colleagues in their comment (in brief: the varimax rotation applied in Carmona et al. 2021 does not affect trait relationships; using a stepwise approach when including traits does not improve the inferences about trait relationships, and adding seed mass does not affect the relationships between the other traits), and showed that the apparent differences between the results of these papers stem from different interpretations of essentially the same trait space.

In any case, our study is not devoted to addressing the discrepancies between these two studies that find contrasting patterns about the coordination between leaf and root economics axes. Rather, we aim to disentangle the coordination among aboveground and belowground size. Accordingly, in the Discussion, we directly address the possible reasons behind the different results related to the inclusion of root size traits in our UPFS and in Weigelt et al. (2021) as follows (lines 234-256):

“Our analyses show that root trait variation is summarized along three independent dimensions of variation, such that adding either root size trait to the RES expands the dimensionality of this space to a similar extent as adding an uncorrelated trait. Interestingly, this aspect aligns with the results of Weigelt et al. (2021), but with a notable divergence. Unlike their study, which found only a weak correlation between aboveground size and rooting depth, our analyses indicate a stronger linkage. We attribute the observed disparity to the imbalance in the representation of woody species between our study and that of Weigelt et al. (2021). Specifically, of the 540 species with common information for plant height and rooting depth in Weigelt et al. (2021), 62.41% were woody and 33.52% herbaceous. By contrast, in our study, out of 1631 species sharing observations for the two traits, 36.54% were woody and 58.86% were herbaceous (around 45%-48% of global vascular plant species are estimated to be woody (FitzJohn et al., 2014). Factors such as variations in water table depth, aridity, and precipitation significantly influence the rooting depth of woody species globally (Tumber-Dávila et al., 2022; Laughlin et al., 2023). In our dataset, when analysed independently, woody species exhibited greater variability in rooting depth compared to herbaceous species, albeit without delineating an independent

axis of variation (Supplementary Fig. 4). The discrepancy in the proportion of woody species between the two studies may potentially result in an amplification in the rooting depth disparities among species showing different growth forms, resulting in different dimensionality among the two spaces. Crucially, our findings demonstrate a robust plant size coordination in the UPFS, unaffected by the specific root size traits or considered growth form, the completeness of traits information, or the phylogenetic relationships of the species. These results point to a strong, consistent coordination between above- and belowground size traits in the global context of the UPFS. Nevertheless, we advocate for future research to focus on differential investments within woody and herbaceous growth forms and to examine the impact that varying proportions of these growth forms have on the outcomes of global syntheses like ours. “

In this latter version of the text, we included a clear statement about the need to disentangle the effects of different proportions of growth forms in the database, which we believe may be a cause of the difference in results between our work and Weigelt et al. (2021).

Line 524-533: It would help the reader if the trait labels would not that strongly overlap in the figure.

R30: Done, we moved the labels to enhance the readability of the figure.

Reviewer #3 (Remarks on code availability):

The code provides a README file and is well commented and explained. I did not run the code.

R31: Thank you!

References:

Bergmann J, Weigelt A, van der Plas F, Laughlin DC, Kuyper TW, Guerrero-Ramirez N, Valverde-Barrantes OJ, Bruelheide H, Freschet GT, Iversen CM, *et al.* 2020. The fungal collaboration gradient dominates the root economics space in plants. *Science Advances* 6: eaba3756.

Bueno CG, Toussaint A, Träger S, Díaz S, Moora M, Munson AD, Pärtel M, Zobel M, Tamme R, Carmona CP. 2023. Reply to: The importance of trait selection in ecology. *Nature* 618: E31–E34.

Capdevila P, Walker TWN, Schrodte F, Caro RCR, Salguero-Gomez R. 2023. Global patterns of plant form and function are strongly determined by evolutionary relationships. : 2023.01.13.523963.

Carmona CP, Bueno CG, Toussaint A, Träger S, Díaz S, Moora M, Munson AD, Pärtel M, Zobel M, Tamme R. 2021. Fine-root traits in the global spectrum of plant form and function. *Nature* 597: 683–687.

Chelli S, Klimešová J, Tsakalos JL, Puglielli G. 2024. Unravelling the clonal trait space: Beyond above-ground and fine-root traits. *Journal of Ecology* 112: 730–740.

Díaz S, Kattge J, Cornelissen JHC, Wright IJ, Lavorel S, Dray S, Reu B, Kleyer M, Wirth C, Colin Prentice I, *et al.* 2016. The global spectrum of plant form and function. *Nature* 529: 167–171.

FitzJohn RG, Pennell MW, Zanne AE, Stevens PF, Tank DC, Cornwell WK. 2014. How much of the world is woody? *Journal of Ecology* 102: 1266–1272.

Funk JL, Larson JE, Blair MD, Nguyen MA, Rivera BJ. 2024. Drought response in herbaceous plants: A test of the integrated framework of plant form and function. *Functional Ecology* 38: 679–691.

Iversen CM, McCormack ML, Powell AS, Blackwood CB, Freschet GT, Kattge J, Roumet C, Stover DB, Soudzilovskaia NA, Valverde-Barrantes OJ, *et al.* 2017. A global Fine-Root Ecology Database to address below-ground challenges in plant ecology. *New Phytologist* 215: 15–26.

Kattge J, Bönisch G, Díaz S, Lavorel S, Prentice IC, Leadley P, Tautenhahn S, Werner GDA, Aakala T, Abedi M, *et al.* 2020. TRY plant trait database – enhanced coverage and open access. *Global Change Biology* 26: 119–188.

Klimešová J, Ottaviani G, Charles-Dominique T, Campetella G, Canullo R, Chelli S, Janovský Z, Lubbe FC, Martínková J, Herben T. 2021. Incorporating clonality into the plant ecology research agenda. *Trends in Plant Science* 26: 1236–1247.

Klimešová J, Tackenberg O, Herben T. 2016. Herbs are different: clonal and bud bank traits can matter more than leaf–height–seed traits. *New Phytologist* 210: 13–17.

Laughlin DC. 2014. The intrinsic dimensionality of plant traits and its relevance to community assembly. *Journal of Ecology* 102: 186–193.

Laughlin DC. 2023. Plant Functional Traits and the Multidimensional Phenotype. In: Laughlin DC, ed. *Plant Strategies: The Demographic Consequences of Functional Traits in Changing Environments*. Oxford University Press, 159–212.

Laughlin DC, Siefert A, Fleri JR, Tumber-Dávila SJ, Hammond WM, Sabatini FM, Damasceno G, Aubin I, Field R, Hatim MZ, *et al.* 2023. Rooting depth and xylem vulnerability are independent woody plant traits jointly selected by aridity, seasonality, and water table depth. *New Phytologist* n/a: 1774–1787.

Legendre P, Legendre L. 2012. *Numerical ecology*. Elsevier.

McCarthy MC, Enquist BJ. 2007. Consistency between an Allometric Approach and Optimal Partitioning Theory in Global Patterns of Plant Biomass Allocation. *Functional Ecology* 21: 713–720.

Polly PD, Lawing AM, Fabre A-C, Goswami A. 2013. Phylogenetic Principal Components Analysis and Geometric Morphometrics. *Hystrix, the Italian Journal of Mammalogy* 24.

Puglielli G, Laanisto L, Poorter H, Niinemets Ü. 2021. Global patterns of biomass allocation in woody species with different tolerances of shade and drought: evidence for multiple strategies. *New Phytologist* 229: 308–322.

Reich PB. 2014. The world-wide ‘fast–slow’ plant economics spectrum: a traits manifesto. *Journal of Ecology* 102: 275–301.

Revell LJ. 2009. Size-Correction and Principal Components for Interspecific Comparative Studies. *Evolution* 63: 3258–3268.

Revell LJ, Harmon LJ. 2022. *Phylogenetic Comparative Methods in R*. Princeton University Press.

Schenk HJ, Jackson RB. 2002. Rooting Depths, Lateral Root Spreads and Below-Ground/Above-Ground Allometries of Plants in Water-Limited Ecosystems. *Journal of Ecology* 90: 480–494.

Smith-Martin CM, Xu X, Medvigy D, Schnitzer SA, Powers JS. 2020. Allometric scaling laws linking biomass and rooting depth vary across ontogeny and functional groups in tropical dry forest lianas and trees. *New Phytologist* 226: 714–726.

Tumber-Dávila SJ, Schenk HJ, Du E, Jackson RB. 2022. Plant sizes and shapes above and belowground and their interactions with climate. *New Phytologist* 235: 1032–1056.

Uyeda JC, Caetano DS, Pennell MW. 2015. Comparative Analysis of Principal Components Can be Misleading. *Systematic Biology* 64: 677–689.

Weigelt A, Mommer L, Andraczek K, Iversen CM, Bergmann J, Bruelheide H, Fan Y, Freschet GT, Guerrero-Ramírez NR, Kattge J, *et al.* 2021. An integrated framework of plant form and function: the belowground perspective. *New Phytologist* 232: 42–59.

Weigelt A, Mommer L, Andraczek K, Iversen CM, Bergmann J, Bruelheide H, Freschet GT, Guerrero-Ramírez NR, Kattge J, Kuyper TW, *et al.* 2023. The importance of trait selection in ecology. *Nature* 618: E29–E30.

Wright IJ, Reich PB, Westoby M, Ackerly DD, Baruch Z, Bongers F, Cavender-Bares J, Chapin T, Cornelissen JHC, Diemer M, *et al.* 2004. The worldwide leaf economics spectrum. *Nature* 428: 821–827.

Manuscript Title: *Aboveground and belowground sizes are aligned in the unified spectrum of plant form and function*

Eleonora Beccari & Carlos P. Carmona

We are grateful for the opportunity to resubmit a revised version of our manuscript to Nature Communications. We have corrected the typos mentioned by Reviewer #3 and formatted the text according to the journal's guidelines. The response text is presented in black font, while the reviewer's comments are in dark orange

Reviewer #3 (Remarks to the Author):

Dear Authors and Editor,

This is the second version of this manuscript that I have reviewed. When I reviewed an earlier version, my overall assessment was positive. I am glad that the authors carefully considered my main comments, especially regarding the debate between Carmona et al. (2021) and Weigelt et al. (2021). I also greatly appreciate the interesting insights provided by the comparison of their original database to an updated version of the TRY database. I thank the authors for their excellent work, and I have only a few minor comments remaining.

R1: Thanks for your work on our manuscript and for your support.

Line 141: Typo in 'Root lateral spread'

R2: Amended

Line 173: 'Lateral root spread' instead of 'Root lateral spread' – please ensure consistent terminologies.

R3: Done.